# DLMM as a lossless one-shot algorithm for collaborative multi-site distributed linear mixed models

Chongliang Luo[1,2], Md. Nazmul Islam[3], Natalie E. Sheils [3], John Buresh[3], Jenna Reps [4], Martijn J. Schuemie[4], Patrick B. Ryan[4], Mackenzie Edmondson [1], Rui Duan [1,5], Jiayi Tong [1], Arielle Marks-Anglin[1], Jiang Bian [6], Zhaoyi Chen[6], Talita Duarte-Salles[7], Sergio Fernández-Bertolín[7], Thomas Falconer[8], Chungsoo Kim [9], Rae Woong Park [9,10], Stephen R. Pfohl[11], Nigam H. Shah [11], Andrew E. Williams [12], Hua Xu[13], Yujia Zhou [13], Ebbing Lautenbach[1,14,15], Jalpa A. Doshi[16,17], Rachel M. Werner[16,17,18], David A. Asch [16,17] & Yong Chen [1✉]

Linear mixed models are commonly used in healthcare-based association analyses for analyzing multi-site data with heterogeneous site-specific random effects. Due to regulations for protecting patients' privacy, sensitive individual patient data (IPD) typically cannot be shared across sites. We propose an algorithm for fitting distributed linear mixed models (DLMMs) without sharing IPD across sites. This algorithm achieves results identical to those achieved using pooled IPD from multiple sites (i.e., the same effect size and standard error estimates), hence demonstrating the lossless property. The algorithm requires each site to contribute minimal aggregated data in only one round of communication. We demonstrate the lossless property of the proposed DLMM algorithm by investigating the associations between demographic and clinical characteristics and length of hospital stay in COVID-19 patients using administrative claims from the UnitedHealth Group Clinical Discovery Database. We extend this association study by incorporating 120,609 COVID-19 patients from 11 collaborative data sources worldwide.

[1] Department of Biostatistics, Epidemiology and Informatics, University of Pennsylvania, Philadelphia, PA, USA. [2] Division of Public Health Sciences, Washington University School of Medicine in St. Louis, St. Louis, MO, USA. [3] Optum Labs, Minnetonka, MN, USA. [4] Janssen Research and Development LLC, Titusville, NJ, USA. [5] Department of Biostatistics, Harvard T.H. Chan School of Public Health, Boston, MA, USA. [6] Department of Health Outcomes and Biomedical Informatics, College of Medicine, University of Florida, Gainesville, FL, USA. [7] Fundacio Institut Universitari per a la recerca a l'Atencio Primaria de Salut Jordi Gol i Gurina (IDIAPJGol), Barcelona, Spain. [8] Department of Biomedical Informatics, Columbia University, New York, NY, USA. [9] Department of Biomedical Sciences, Ajou University Graduate School of Medicine, Suwon, Republic of Korea. [10] Department of Biomedical Informatics, Ajou University School of Medicine, Suwon, Republic of Korea. [11] Stanford Center for Biomedical Informatics Research, Stanford, CA, USA. [12] Institute for Clinical Research and Health Policy Studies, Tufts University School of Medicine, Boston, MA, USA. [13] School of Biomedical Informatics, The University of Texas Health Science Center at Houston, Houston, TX, USA. [14] Division of Infectious Diseases, Department of Medicine, Perelman School of Medicine, University of Pennsylvania, Philadelphia, PA, USA. [15] Center for Clinical Epidemiology and Biostatistics, Perelman School of Medicine, University of Pennsylvania, Philadelphia, PA, USA. [16] Division of General Internal Medicine, University of Pennsylvania, Philadelphia, PA, USA. [17] Leonard Davis Institute of Health Economics, Philadelphia, PA, USA. [18] Cpl Michael J Crescenz VA Medical Center, Philadelphia, PA, USA. ✉email: ychen123@upenn.edu

Integrating data from multiple sites can increase statistical power and generalizability[1]. The recent development of clinical research networks (CRNs) have established networks of researchers and databases for multi-site, large-scale analyses[2]. For example, in the United States, the National Patient-Centered Clinical Research Network (PCORnet) is a large, highly representative electronic data infrastructure, covering 337 hospitals with data from 80 million patients[3–5]. PCORnet was established a decade ago to improve comparative effectiveness research, patient-centered outcome research, and pragmatic trials. The international Observational Health Data Sciences and Informatics (OHDSI) network[6] has accumulated more than half a billion patient records from 19 different countries worldwide, with around 300 million patient records within the United States. Other research networks include the ACT Network (Accrual to Clinical Trials)[7], the Food and Drug Administration's Sentinel Initiative[8], and TriNetX[9], among others. In response to the COVID-19 pandemic, there have been great efforts to build national COVID cohorts and improve data infrastructure[10]. Significant examples include the National COVID Cohort Collaborative (N3C) and the Consortium for Characterization of COVID-19 by EHR (4CE), which is an international consortium of 96 hospitals across five countries[11].

Research networks are vital for advancing clinical research and responding to the COVID-19 crisis. For example, Burn et al.[12] analyzed the characteristics of adults hospitalized with COVID-19 and compared them with influenza patients based on the OHDSI network. However, while multi-site studies via CRNs are promising, they present several key challenges. First, most CRNs convert their data to a common data model (CDM) while retaining the data at their individual clinical sites. Sensitive individual patient data (IPD) including patient demographics, diagnoses, and treatments usually cannot be shared across networks under privacy regulations such as the Health Insurance Portability and Accountability Act (HIPAA) in the United States or the General Data Protection Regulation (GDPR) in the E.U. The second challenge is the requirement of iterative communications in the existing federated/distributed learning framework[13,14]. Although these federated learning algorithms allow modeling on distributed datasets without sharing patient-level data, they typically require iterative processes, resulting in multiple communications among data contributors that add effort and delay. Implementation of these algorithms usually requires extra computation infrastructures across networks[15] to allow iterative communications of model parameter estimates. Thirdly, the data from across sites are often heterogeneous due to difference in patient characteristics and other site-level variations. Ignoring this heterogeneity could lead to biased estimates and misleading predictions from the analyses.

The above challenges add to the timely responses to the COVID-19 and other time sensitive research. To address these challenges, we propose an algorithm for performing distributed linear mixed models (DLMMs) when integrating data from multiple sites. The linear mixed model (LMM) is an extension of the linear model and allows for modeling site-specific effects of covariates on the outcome. The LMM assumes site-specific random effects in addition to common fixed effects. Random effects help us quantify the systematic deviations, e.g., administrative differences across sites, from the overall population-level effects which are characterized by the common fixed effects. The estimation of common fixed effects enables general knowledge discovery, while the estimation of random effects allows quantification of between-site heterogeneity and site-specific predictions.

Computation of the LMM requires maximizing the profile likelihood[16], whose calculation cannot be distributed to different sites using the traditional technique of matrix decomposition[17,18]. Regular federated learning algorithms[13,14,19] require the iterative transmission of aggregated data (AD) and therefore are not communication efficient. Instead, our proposed algorithm requires each site to contribute AD only once, but achieves identical results (i.e., identical effect size and standard error estimates) as the analysis from the pooled IPD data. This DLMM algorithm has four properties:

1. Privacy-preserving: Only requires sharing of AD
2. Communication-efficient: Only requires one round of communication
3. Heterogeneity-aware: Allows for site-specific effects
4. Lossless: Achieves identical results (i.e., identical effect size and standard error estimates) as the analysis from the pooled IPD data

See Fig. 1 for a comparison of the existing and the proposed approaches for analyzing multi-site data. We note that, generally, an algorithm being lossless has to sacrifice certain properties of the algorithm, such as the privacy protection or communication-efficiency. However, there are rare exceptions algorithms can be both lossless and privacy-protected. One such example is the lossless distributed algorithm for linear regression[18], being lossless, communication-efficient and privacy-protected. Our proposed algorithm inherits such unique property of the distributed linear regression for being both lossless and privacy-preserving, yet being able to account for between-site heterogeneity.

In this paper, we use the length of stay (LOS) of hospitalized COVID-19 patients as an illustrating example. LOS has a direct implication on hospital capacity, as a commonly used and easily measured indicator of inpatient health care utilization. LOS is an important outcome for the evaluation of clinical interventions[20].

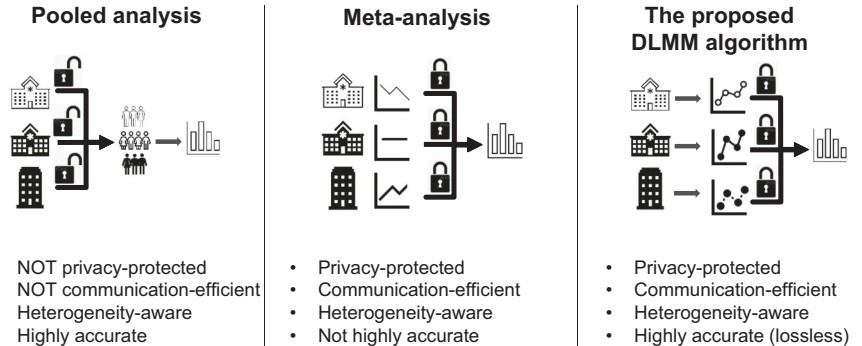

**Pooled analysis**
- NOT privacy-protected
- NOT communication-efficient
- Heterogeneity-aware
- Highly accurate

**Meta-analysis**
- Privacy-protected
- Communication-efficient
- Heterogeneity-aware
- Not highly accurate

**The proposed DLMM algorithm**
- Privacy-protected
- Communication-efficient
- Heterogeneity-aware
- Highly accurate (lossless)

**Fig. 1 Comparison of approaches for multi-site analysis with heterogeneity across sites.** Pooled analysis: data analysis based on the pooled individual patient data; Meta-analysis: statistical analysis by combining the effect size and standard error estimates from different sites; The proposed DLMM algorithm: the distributed linear mixed model algorithm.

Identifying factors associated with variation in LOS can lead to a better understanding of how to improve clinical care and economic efficiency. In addition, LOS is an ideal outcome in our illustration to study heterogeneous outcomes in real world data. For example, Rees and colleagues[21] found that the LOS of COVID-19 hospitalization in China is longer than in other countries, possibly because China has stricter discharge criteria. When studying the association between LOS and such patient characteristics as age, gender, Charlson Comorbidity Index (CCI), and history of cancer, heart disease, hypertension, hyperlipidemia, and diabetes, the impact of these variables on the LOS of COVID-19 hospitalization may differ across sites[22], representing a common situation of heterogeneity in not only the baseline (i.e., the mean LOS across different sites), but also the associations between these risk factors and the LOS outcome.

In the Results section, we first formulate the LMM and the proposed DLMM algorithm, and then validate the above properties of the DLMM algorithm using data from the UnitedHealth Group Clinical Discovery Portal. We analyze the pooled data from 538 hospitals using regular LMM and compare these results with those from the DLMM algorithm applied to the same data without allowing IPD sharing. We then apply the DLMM algorithm to analyze the association between LOS of COVID-19 hospitalization with patients' demographic, admission, and clinical characteristics using the electronic health records (EHR) and claims from 11 databases across three countries (US, Spain, and South Korea) with a total of 120,609 COVID-19 patients.

## Results

**Distributed linear mixed model and the lossless property.**
Linear mixed models with site-level random effects can be used to account for the heterogeneous effects of covariates on a continuous outcome. Assume for the $j$th patient at the $i$th site, $y_{ij}$ is the continuous outcome, $x_{ij}$ is the $p$-dimensional covariate vector, $\beta$ is the vector of fixed effects, $z_{ij}$ is the $q$-dimensional covariate vector, $u_i$ is the $q$-dimensional random effect, and $\epsilon_{ij}$ is the random error.

$$y_{ij} = x_{ij}^T\beta + z_{ij}u_i + \epsilon_{ij}, i = 1, ..., K, j = 1, ..., n_i, \quad (1)$$

where $u_i \sim N(0, V), \epsilon_{ij} \sim N(0, \sigma^2)$. The random effects covariates $z_{ij}$ can be part or all of $x_{ij}$, or constant 1 if representing a random intercept only. The random effect covariance matrix $V$ can have certain structures with unknown parameters. For instance, we can assume the random effects are independent, i.e., $V = diag(\sigma_1^2, ..., \sigma_q^2)$. These parameters (e.g., variance components) and the fixed effects $\beta$ are usually estimated by maximum likelihood (ML) or restricted maximum likelihood (REML) estimation[17].

For the scenario in which IPD data are distributed at multiple sites and pooling is restricted, we have developed a distributed algorithm to fit the LMM in a privacy-preserving and communication-efficient manner. Figure 2 demonstrates how the proposed DLMM algorithm works. The algorithm requires only aggregated data $S_i^X = X_i^TX_i, S_i^{Xy} = X_i^Ty_i, s_i^y = y_i^Ty_i$ and sample size $n_i$ from the $i$th site to reconstruct the likelihood for fitting the model. Notice the communication of aggregated data is required only once, and the reconstructed likelihood leads to identical results compared to the pooled analysis that requires IPD from all sites. See the Methods Section for more details of the algorithm.

We demonstrate the utility and lossless property of the DLMM method by studying the association of COVID-19 hospitalization LOS with patients' demographic and clinical characteristics, using the medical claims data from the UnitedHealth Group (UHG)

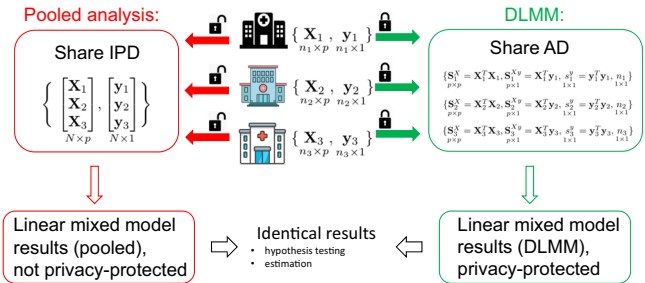

**Fig. 2 Schematic overview of the proposed algorithm for distributed linear mixed model (DLMM).** The linear mixed model takes into account the heterogeneity of the effect of the covariates $X$ (e.g., patients' characteristics) on the continuous outcome $y$ (e.g., COVID-19 hospitalization length of stay) across sites. The proposed distributed algorithm achieves identical results as pooling the individual patient data (IPD) from all sites, by requiring only aggregated data (AD) $S_i^X, S_i^{Xy}, s_i^y$ and sample size $n_i$ from the $i$th site. The DLMM algorithm does not depend on a "leading site" and any site can conduct the analysis given the aggregated data are available.

Clinical Discovery Portal. Since we can access all the patient-level data, comparing the results of pooled analysis and distributed analysis can demonstrate the lossless property of the proposed method. We identified patients who were admitted as inpatients to a hospital with a primary diagnosis of COVID-19 between January 1, 2020 and September 30, 2020. The data were collected from $K = 538$ sites (i.e., hospitals) and the total number of patients is $N = \sum_{i=1}^{538}n_i = 47,756$. The detailed inclusion criteria are in the Supplementary Figs. 1 and 2. LOS was treated as a continuous outcome. The variation in LOS across the sites is shown in Fig. 3a. The demographic characteristics include age, gender, and race, and the clinical characteristics include a history of cancer, chronic obstructive pulmonary disease (COPD), heart disease, hypertension, hyperlipidemia, diabetes, kidney disease, and obesity. CCI score is also included as a measure of the overall patient's health state; the higher the score, the worse the health state is. We provide the details of the definition of the characteristics in the Supplementary Table 1.

We compare the result of the pooled analysis and the distributed algorithm in Fig. 3. Specifically, the estimation of the fixed effects, their standard errors, and the variance components are shown to be identical by the pooled analysis or the distributed algorithm. The random effects of obesity, diabetes, kidney disease, and CCI categories are significant. The estimated best linear unbiased predictors (BLUPs) (where "best" refers to the minimal variance of the estimate among all the unbiased linear estimates) of the random effects and their variances are presented in the Supplementary Fig. 3, comparing the pooled analysis and the distributed algorithm.

Figure 4 shows the forest plots of fixed effects estimation and BLUPs of the random effects at a specific site. Older age ($\geq 80$), male gender, and diagnosis of diabetes, kidney disease, or obesity, as well as higher CCI, are shown to be significantly associated with longer COVID-19 hospitalization. Non-Hispanic white (NHW) race, cancer, and hyperlipidemia are significantly associated with shorter COVID-19 hospitalization. These results agree with previous findings[21,23–25].

**International COVID-19 hospital LOS study.** We further demonstrated the applicability of the proposed DLMM algorithm by investigating the association of COVID-19 hospitalization LOS with patient characteristics, using the EHR and medical claims data from 11 data sources within and outside of the United States.

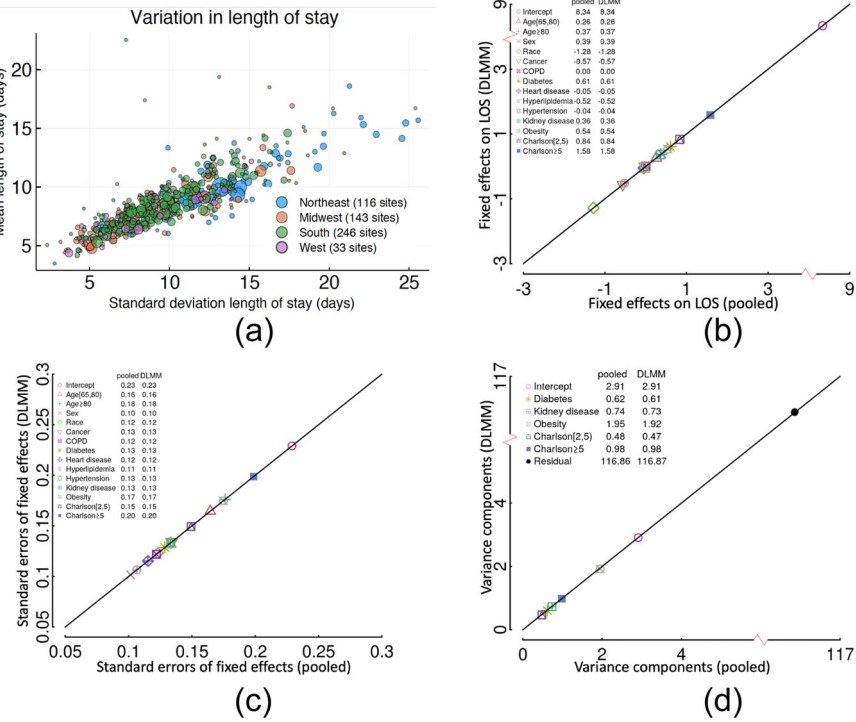

**Fig. 3 The lossless property of the DLMM algorithm illustrated by the COVID-19 LOS study using the UHG data. a** The mean and standard deviation of length of stay of 47,756 hospitalized COVID-19 patients from 538 hospitals. The data were collected from a single large U.S. insurer via the UHG Clinical Discovery Portal and are separated into their respective hospital sites to illustrate the algorithm. The area of each dot is proportional to the number of patients at that hospital, and the color represents the region. **b** Fixed effects estimation of linear mixed model by the proposed DLMM algorithm vs. the pooled analysis. **c** Fixed effects' standard error estimation of linear mixed model by the proposed DLMM algorithm vs. the pooled analysis. **d** Variance components estimation of linear mixed model by the proposed DLMM algorithm vs. the pooled analysis.

These data sources are shown in Fig. 5. The detailed description is in the Supplementary Notes.

We categorize the UHG Clinical Discovery Portal data as four "sites" based on geographical area, i.e., Northeast (UHG.NE), South (UHG.S), West (UHG.W) and Midwest (UHG.MW). The number of sites is thus $K = 14$, and the total number of patients is $N = \sum_{i=1}^{K} n_i = 120,609$. The detailed inclusion and exclusion criteria are in the Supplementary Notes. In addition to the demographic and clinical characteristics defined previously, we also included the admission date (categorized as Q1, Q2, or Q3, i.e., admission in the first, second, or third quarter of 2020, respectively). Race is excluded from the covariates, since race information is missing from some data sources. A data summary of the collaborative sites is in Supplementary Tables 3 and 4.

The random effects of all covariates are forward selected by the likelihood ratio test. All covariates except obesity are tested as having significant random effects (see Supplementary Table S4). The results of the estimated fixed effects and prediction (i.e., estimated BLUPs) of the site-specific random effects are presented in Figs. 6 and 7.

Below are the risk factors with significant fixed effects (p-value < 0.001), as shown in Fig. 6.

- Age and gender are significant risk factors for increased LOS. Age [65, 80) is associated with 0.96 days (95% CI = 0.44–1.48) longer LOS, compared to age [18, 65); and male gender is associated with 0.58 days (95% CI = 0.24–0.92) longer LOS, compared to female gender.
- CCI is a significant risk factor for LOS. CCI [2,5) and 5+ are associated with 1.42 days (95% CI = 0.58–2.27) and 2.68 days (95% CI = 1.16–4.21) longer LOS respectively, compared to CCI [0, 2).

- Hospitalizations in the later time intervals are associated with shorter LOS. Hospitalizations in the second and third quarter are associated with 3.79 (95% CI = 3.03–4.55) and 6.26 (95% CI = 5.23–7.30) days shorter LOS, compared to the first quarter.
- Among those comorbidity conditions, obesity is a significant risk factor of LOS, associates with 0.37 (95% CI = 0.21–0.52) days longer LOS.

Regarding the site-specific random effects, below are comparisons among sites, as shown in Fig. 7. Compared to other sites,

- The baseline LOS (i.e., the intercept) is longer in in HIRA COVID, and shorter in SIDIAP.
- Age 80+ is associated with shorter LOS in CUIMC, and age 65-80 and male gender are associated with longer LOS in CCAE.
- Higher CCI score is associated with longer LOS in CUIMC.
- CUIMC and Optum EHR have larger effects of admission in the second quarter (i.e., longer LOS compared to the first quarter), and UHG.NE has smaller effects of admission in the second and third quarter.
- History of COPD, kidney and heart diseases are associated with longer LOS in Optum COVID, and history of hypertension, heart disease and diabetes are associated with shorter LOS in CUIMC.

## Discussion
Special care must be taken with health data in order to preserve patient privacy. Anonymizing data while preserving features that are important for understanding an individual's health is nontrivial. In addition, large, representative datasets are especially

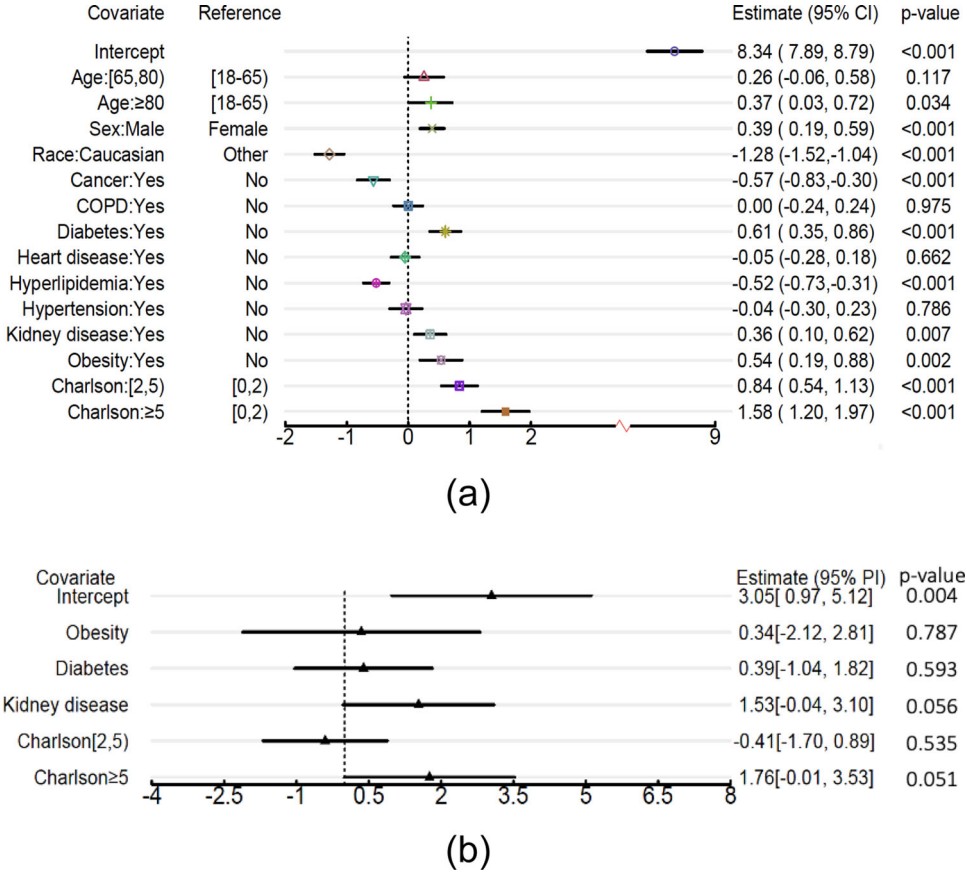

**Fig. 4 Distributed linear mixed model estimation of the COVID-19 LOS study using the UHG data.** The UnitedHealth Group (UHG) Clinical Discovery Portal medical claims data contains $n = 47,756$ independent patients from 538 sites (i.e., hospitals). **a** Fixed effects of demographic and clinical characteristics on COVID-19 hospitalization length of stay. A vertical reference line is drawn for convenience in comparison. Reported are the estimated effect sizes and 95% confidence intervals. **b** BLUPs and 95% prediction intervals for random effects corresponding to a site located in the south region. The 95% confidence (or prediction) intervals are presented as point estimate ±1.96 * standard error estimate. The corresponding p-values are based on two-sided Wald tests.

scarce. Distributed models deal with the privacy issue by requiring that only summary level statistics are shared. The one-shot model presented here requires only the aggregated data of a $p \times p$ matrix, $p$-dimensional vector, and sample size be sent once, where $p$ is the number of risk factors in the association analysis. This allows the data to remain completely protected by eliminating the need for data pooling at a central source. By considering a large, more diverse sample from multiple sites, we expect a more robust estimation, which benefits general knowledge discovery[26,27]. While we use the COVID-19 hospitalization LOS study as an illustrative example, this distributed linear mixed model algorithm can be applied to any continuous outcome where potential heterogeneity exists for the effects of covariates across sites.

Studying LOS heterogeneity across sites could help understand and potentially predict future LOS, which is instrumental for relieving the burden of healthcare systems during the pandemic resurgence. In light of this, we are designing an online portal for further collaboration. As the pandemic continues, more hospitalization data may be collected from additional countries/areas. Therefore, it is important to update the association study in a timely manner. With this online portal, any data source can contribute aggregated data to update the estimation, and more importantly, get a prediction of the site-specific effects of the characteristics on COVID-19 LOS.

As suggested by a reviewer, the linear mixed model is closely connected with the (random-effects) meta-analysis, as they both

assume the association effects are random and can shrink site-specific (or study-specific) estimation which benefits prediction performance. A comparison of our LMM (or equivalently DLMM) and the random-effects meta-analysis for the LOS study is demonstrated in Supplementary Fig. 5. The results show that the estimation of common fixed-effects and site-specific random effects (i.e., BLUPs) are similar but not identical. However, such difference depends on various factors, such as the number of patients per site, the ratio between the within-site heterogeneity and the between-site heterogeneity, and the number of sites. Meta-analysis-based model aggregation is extensively studied in the literature for prediction purposes[28–30]. A comprehensive comparison between LMM and meta-analysis is however beyond the scope of this paper. We also note that in the setting of modeling interactions, caution should be taken in the formulation of the regression model in LMM. Through various proposed formulations that distinguish between-study information from the within-study information[31,32], the aggregation bias can be avoided.

Our DLMM algorithm is considered privacy-preserving as it only requires one-shot communication of aggregated data from collaborative sites, and the aggregated data are only shared within collaborators who participate in the study. However, our aggregated data release mechanism has not been rigorously studied to meet privacy-preserving criteria such as $k$-anonymity or differential privacy[33–35]. Specifically, the $k$-anonymity property protects against the risk of re-identification[33], which arises from

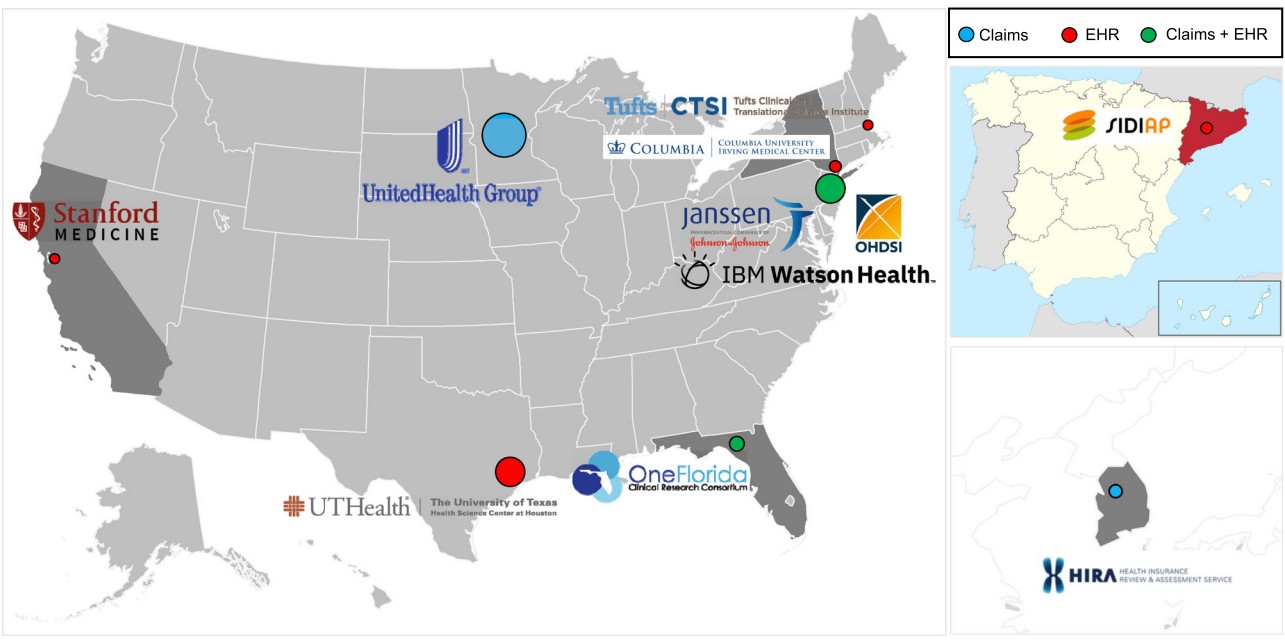

**Fig. 5 Map of the international data sources of the LOS for COVID-19 hospitalization.** The headquarters of data sources are marked. The CUIMC, STARR, OneFlorida, and TRDW cover parts of New York, California, Florida, and Massachusetts. The UHG, OHDSI (CCAE, MDCR, Optum EHR), and Optum COVID cover multiple states in the United States. The SIDIAP and HIRA COVID cover Spain-Catalonia and South Korea, respectively. The circle size represents data set size.

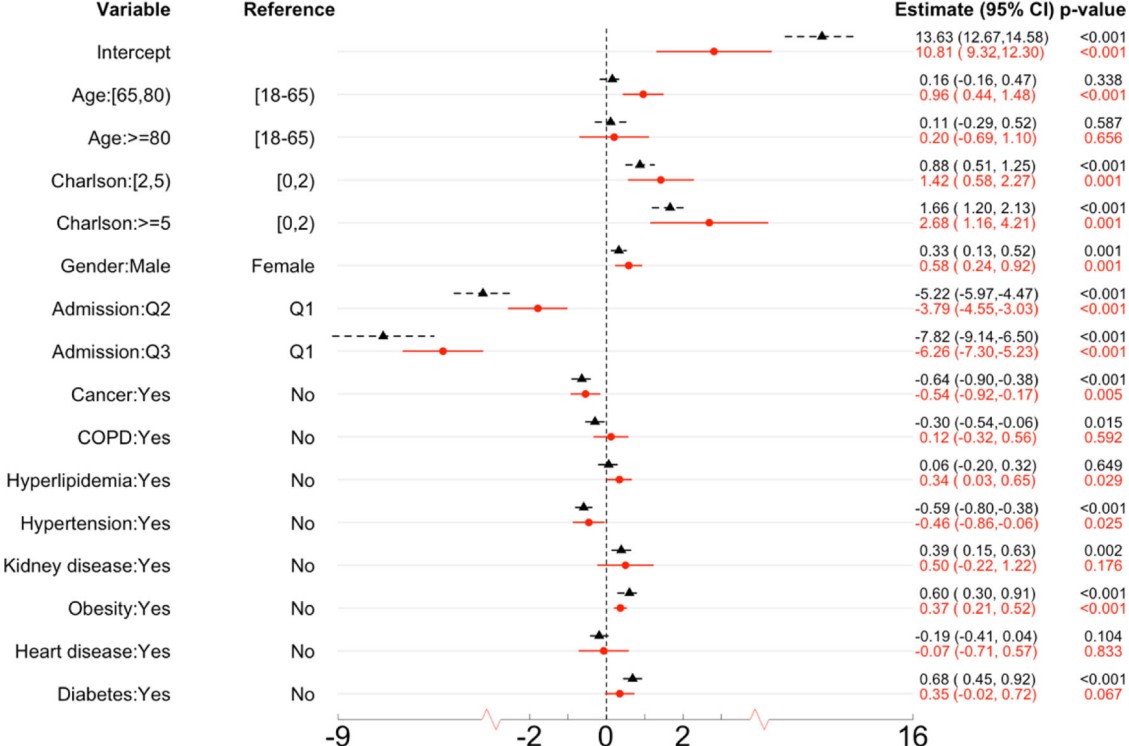

**Fig. 6 The estimated fixed effects and 95% confidence intervals in the COVID-19 LOS study.** The dashed and black CIs are the results using only the 4 UHG sites (the UHG database contains $n = 47,756$ patients and is divided into four sites, i.e., Northeast, South, Midwest, and West), the solid and red CIs are the results using all 14 sites with $n = 120,609$ patients. The results are obtained by using the DLMM algorithm to integrate aggregated data from the multiple sites. The 95% CIs are presented as point estimate ±1.96 * standard error estimate. The corresponding p-values are based on two-sided Wald tests.

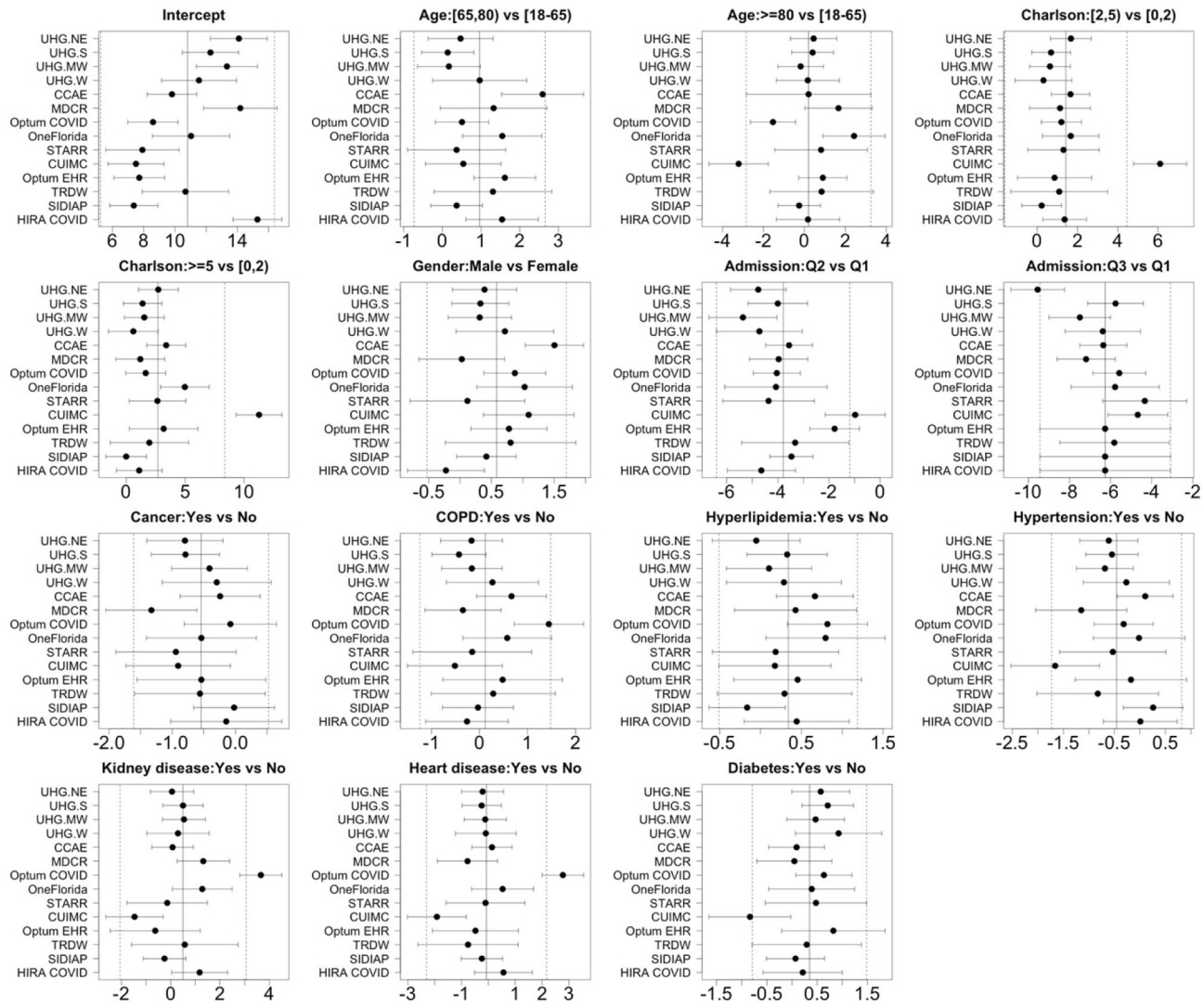

**Fig. 7 The estimated site-specific effects and 95% prediction intervals in the COVID-19 hospitalization LOS study.** The study uses $n = 120,609$ patients from the 14 sites. The site-specific effect is defined as fixed effect plus random effect, and a 95% prediction interval is point estimate ±1.96 * standard error estimate.

linking potential quasi-identifiers (i.e., combinations of patient's characteristics in our study) to external sources[33]. In DLMM aggregated data, if all the cell counts are not sparse, the algorithm can potentially meet the $k$-anonymity requirement. In future collaborations using DLMM, we suggest data contributors review the aggregated data to avoid sparse cells (e.g., no cell count less than 5) before sending them to other sites. We will quantify the risk of privacy leaking more rigorously, and enhance our DLMM algorithm via techniques such as differential privacy and multi-party homomorphic encryption[36] in the future. On the other hand, when reporting the estimated BLUPs for site-specific random effects, caution must be exercised if there is sensitive information specific to sites that could be inferred from the result. We thus suggest the estimated BLUPs not be disclosed if the covariates are sensitive characteristics and write this in the protocol when initiating a collaborative project using the DLMM algorithm in the future. Also, the privacy regulation of releasing aggregated data could vary across countries and data providers. The disclosure of aggregated data in the DLMM algorithm needs to meet the local privacy requirement.

While we are able to integrate data sets from many sites, it is possible that some data sets overlap with one another. For example, the data from STARR/CUIMC/SIDIAP/HIRA COVID are independent, yet data in the UHG Clinical Discovery Portal cover all patients from a single large insurance provider across multiple states and could overlap with other datasets, e.g., CCAE, MDCR, and Optum COVID. As these other datasets include all patients regardless of insurance provider, we estimate a 14% overlap of the UHG data, consistent with their market share. Moreover, while all collaborative sites are able to contribute aggregated data from a common data model, it is difficult to obtain data of equal quality from all sites. For example, due to data recording limitations, missing values exist for some factors (e.g., race) at some sites, e.g., UHG, CCAE, MDCR, and SIDIAP. Although we defined the clinical conditions consistently (e.g., using the OHDSI common data model), the diagnosis of clinical conditions in medical claims and EHR data from various systems and countries are not necessarily consistent. Specifically, the diagnosis of COVID-19 in UHG claims data is based on primary or secondary diagnosis, with 90-95% of patients have a primary diagnosis of COVID-19 at the time admission. However, in the OHDSI network, not every data partner has primary/secondary diagnosis specified in their data, thus another definition was adopted, see the Appendix for details. In addition, some clinical risk factors that could be associated with LOS (e.g., CT scan, fever, and onset time[22]) are unavailable from EHR or claims data.

Death is also poorly captured in some databases (e.g., CCAE, MDCR, and HIRA COVID) and should therefore not be ruled out as a competing risk.

We treated LOS as a continuous outcome, mainly for the purpose of illustrating the proposed distributed algorithm. It would be more reasonable to model LOS as a count outcome via the generalized linear mixed model (GLMM) framework, e.g., Poisson regression with mixed effects, to account for the between-site heterogeneity. Other important outcomes, such as mortality, can also be modeled within the GLMM framework. While GLMM is used in applications such as hospital profiling using multi-center data[37], there are few distributed algorithms for privacy-preserving estimation. This is partially because the estimation of GLMM is computationally intensive and its likelihood function is analytically intractable. One strategy of developing distributed algorithms for GLMM is to use the penalized quasi likelihood[38] which can be built upon our DLMM algorithm; see our dPQL algorithm and its application to hospital profiling without sharing patient-level data[39]. Regarding scalability, our DLMM algorithm has great scalability in terms of large number of sites and large number of patients per site. However, in the presence of high dimensional features (i.e., large $p$), the current algorithm will require sharing of $p \times p$ dimensional matrices, which may be too large to be transferred across sites due to privacy concerns. As a result, studies that involve a large number of predictors/features (e.g., a large-scale genomic study) are not suitable for the proposed method. Extension of DLMM algorithm to improve the scalability on large number of features remains an important area for future research. Lastly, federated learning methods have gained a great deal of attention in many clinical settings in recent years[40]. We also consider the proposed method as one variation of federated learning based on a specific statistical model, i.e., LMM. The LMM holds the promise of flexibility and interpretability of regression coefficients, which are particularly suitable for epidemiological studies. Traditional federated learning models have a focus on prediction, whilst the LMM model in our analyses of LOS outcome focused on quantifying associations of risk factors, which is commonly used in biomedical researches. In the future, we plan to investigate communication-efficient federated learning algorithms in distributed research network settings, which is a much-needed area for new methods.

## Methods

**The multi-site COVID-19 hospitalization LOS data sets.** Data were extracted and included in the study if the patient had an inpatient visit between January 2020 and September 2020 satisfying

- Age 18 years or older.
- A COVID-19 diagnosis or positive test recorded up to 21 days prior to the visit or during the visit.
- Been active in the database for 6 months or more prior to the inpatient visit.
- Did not have a discharge status of "expired" prior to September 30, 2020.

A project protocol was created to implement the study https://github.com/ohdsi-studies/DistributedLMM. In total 11 databases participated in the study and shared the aggregated data results. The description and IRB approval or waiver are listed below. The written informed consents for all the data were waived by the corresponding IRBs.

The UHG data: A database of medical claims and hospitalizations from a national claims data warehouse in the United States. Because no identifiable protected health information was extracted or accessed during the course of the study, and all data were accessed in compliance with the HIPPA rules, IRB approval or waiver of authorization was not required. The official exemption by the UHG IRB is also available.

The OneFlorida data: A CRN that contains robust and longitudinal patient-level linked EHR and claims data from ~15 million Floridians from 12 unique healthcare organizations. At OneFlorida, the access to the HIPAA Limited Data Set was reviewed by the University of Florida Institutional Review Board under

IRB202001831. The analysis was run locally at the University of Florida and only summary statistics were shared.

The Stanford Medicine Research Data Repository (STARR): An EHR database of approximately three million patients from Stanford Hospitals and Clinics and the Lucile Packard Children's Hospital in the United States. The analysis had institutional review board approval for using de-identified data, and thus was determined not to be human subjects research and informed consent was not deemed necessary.

Columbia University Irving Medical Center Data Warehouse (CUIMC): Columbia University EHR database contains records from hospitals in New York City. The analysis had institutional review board approval for using de-identified data, and thus was determined not to be human subjects research and informed consent was not deemed necessary.

IBM MarketScan Commercial Database (CCAE): A database of health insurance claims from large employers and health plans who provide private healthcare coverage to employees, their spouses, and dependents in the United States. The patients are younger than 65. The aggregated data was reviewed by the New England Institutional Review Board (IRB) and were determined to be exempt from broad IRB approval, as this research project did not involve human subject research.

IBM MarketScan Medicare Supplemental Database (MDCR): A database of health insurance claims representing retirees (aged 65 or older) in the United States with primary or Medicare supplemental coverage through privately insured fee-for-service, point-of-service, or capitated health plans. The aggregated data was reviewed by the New England Institutional Review Board (IRB) and were determined to be exempt from broad IRB approval, as this research project did not involve human subject research.

Optum de-identified Electronic Health Record Dataset (Optum EHR): A database of electronic healthcare records for patients in the United States. The aggregated data was reviewed by the New England Institutional Review Board (IRB) and were determined to be exempt from broad IRB approval, as this research project did not involve human subject research.

Optum COVID data: The data are sourced from Optum's longitudinal EHR repository derived from more than 700 hospitals and 7000 clinics, including patients who have documented clinical care from January 2007 through to the most current monthly data release with a documented diagnosis of COVID-19 or acute respiratory illness after February 1, 2020 and/or documented COVID-19 testing. This dataset is fully de-identified and research using this dataset is not qualified as human subject research per UTHealth IRB.

Tufts Medical Center Research Data Warehouse (TRDW): EHR database containing records from Tufts Medical Center, Tufts Children's Hospital, and associated primary and tertiary care clinics fused with oncology data from the Tufts MC Tumor Registry, and death data from the Massachusetts State Registry of Vital Records and Statistics. The analysis had institutional review board approval for using de-identified data, and thus was determined not to be human subjects research and informed consent was not deemed necessary.

The Information System for Research in Primary Care (SIDIAP): An EHR database containing primary care records partially linked to inpatient data representing 80% of the general population in Spain-Catalonia. The use of SIDIAP database was approved by the SIDIAP Scientific Committee and the IDIAPJGol Clinical Research Ethics Committee.

Health Insurance and Review Assessment COVID database (HIRA COVID): A national health insurance claims database in South Korea including all patients who are suspected or confirmed as COVID-19. The analysis had institutional review board approval for using de-identified data, and thus was determined not to be human subjects research and informed consent was not deemed necessary.

**Linear mixed model.** Due to the heterogeneity of data across sites, the effects of the covariates on the outcome among sites in the linear regression model may not always be the same[41]. Thus, a linear mixed model is often used. With the notations in the Results section and model in (1), the log-likelihood of LMM using all the data is

$$L(\beta, \sigma^2, V) = -\frac{1}{2}\sum_{i=1}^{K}\{\log|\Sigma_i| + (y_i - X_i\beta)^T\Sigma_i^{-1}(y_i - X_i\beta)\}, \quad (2)$$

where $X_i$ and $y_i$ are the covariate matrix and the outcome vector of the $i^{th}$ site respectively, $|.|$ is the matrix determinant and $\Sigma_i = \Sigma_i(\sigma^2, V) = Z_iVZ_i^T + \sigma^2 I_{n_i}$.

The maximum likelihood estimation can be further simplified by profiling out $\beta$ and $\sigma^2$ from (2). Denote $\Theta = V/\sigma^2$. Given $\Theta$, the estimation of $\beta$ and $\sigma^2$ are

$$\widetilde{\beta}(\Theta) = \left(\sum_{i=1}^{K} X_i^T \Gamma_i^{-1} X_i\right)^{-1}\left(\sum_{i=1}^{K} X_i^T \Gamma_i^{-1} y_i\right), \quad (3)$$

$$\widetilde{\sigma}^2(\Theta) = \frac{1}{N}\sum_{i=1}^{K}(y_i - X_i\widetilde{\beta}(\Theta))^T\Gamma_i^{-1}(y_i - X_i\widetilde{\beta}(\Theta)), \quad (4)$$

where $\Gamma_i = \Gamma_i(\Theta) = Z_i\Theta Z_i^T + I_{n_i}$. Thus, the profile log-likelihood with respect to only $\Theta$ is

$$L_p(\Theta) = -\frac{1}{2}\sum_{i=1}^{K}\{n_i\log\widetilde{\sigma}^2(\Theta) + \log|\Gamma_i| + (y_i - X_i\widetilde{\beta}(\Theta))^T\Gamma_i^{-1}(y_i - X_i\widetilde{\beta}(\Theta))\}, \quad (5)$$

---

**Box 1 | Pseudo-code of the distributed linear mixed model algorithm**

**1**. In site $i = 1, ..., K$, calculate and share $S_i^X = X_i^T X_i$, $S_i^{Xy} = X_i^T y_i$, $s_i^y = y_i^T y_i$ and sample size $n_i$.

**2**. Perform the likelihood ratio test for the significance of random effects of each covariate by Eq. (10).

**3**. With the significant random effects identified by the above step, reconstruct the profile log-likelihood Equation (5) or the restricted profile log-likelihood Eq. (6), obtain the estimate $\hat{\Theta}$.

**4**. Obtain $\hat{\beta} = \tilde{\beta}(\hat{\Theta})$ and $\hat{\sigma}^2 = \tilde{\sigma}^2(\hat{\Theta})$ by Eqs. (3) and (4).

**5**. Calculate the variance of the estimated fixed effects $\hat{\beta}$ by Eq. (7).

**6**. Calculates the BLUPs of the random effects in each site by Eq. (11).

---

and the restricted profile log-likelihood is

$$L_r(\Theta) = L_p(\Theta) - \frac{1}{2}\sum_{i=1}^{K}\{\log|X_i^T \Gamma_i^{-1} X_i| - n_i \log \tilde{\sigma}^2(\Theta)\}, \quad (6)$$

The ML or REML estimate of $\Theta$ can be obtained by maximizing (5) or (6). The estimates of $\beta$ and $\sigma^2$ can be subsequently obtained by (3) and (4). We denote these estimates as $(\hat{\beta}, \hat{\sigma}^2, \hat{\Theta})$. Thus, the variance of the estimated fixed effects $\hat{\beta}$ is

$$V(\hat{\beta}) = \hat{\sigma}^2 \left(\sum_{i=1}^{K} X_i^T \Gamma_i(\hat{\Theta})^{-1} X_i\right)^{-1}, \quad (7)$$

or the sandwich estimator

$$\left(\sum_{i=1}^{K} X_i^T \Gamma_i(\hat{\Theta})^{-1} X_i\right)^{-1} \left\{\sum_{i=1}^{K} X_i^T \Gamma_i(\hat{\Theta})^{-1}(y_i - X_i\hat{\beta})(y_i - X_i\hat{\beta})^T \Gamma_i(\hat{\Theta})^{-1} X_i\right\}$$
$$\left(\sum_{i=1}^{K} X_i^T \Gamma_i(\hat{\Theta})^{-1} X_i\right)^{-1}.$$

**Distributed linear mixed model**. It is apparent from (5) that there is no closed-form estimation for LMM. Thus, unlike in the ordinary linear model[18], LMM estimation is not trivial as computation cannot be distributed to each site in a lossless fashion. Fortunately, with some linear algebra, we can disentangle the data $(y_i, X_i)$ and the parameters $\Theta$ in $|\Gamma_i|$ and $\Gamma_i^{-1}$ and thus reconstruct the profile log-likelihood (5) without communicating IPD. Specifically, we utilize the Woodbury matrix identity[42] to obtain

$$\Gamma_i^{-1} = I_{n_i} - Z_i(\Theta^{-1} + Z_i^T Z_i)^{-1} Z_i^T, \quad (8)$$

and the matrix determinant lemma[43] to obtain

$$|\Gamma_i| = |I_q + Z_i^T Z_i \Theta|, \quad (9)$$

where $I_q$ is the $q \times q$ identity matrix. The proposed DLMM algorithm requires the $i^{th}$ site to communicate

- $p \times p$ matrix $S_i^X = X_i^T X_i$,
- $p - dim$ vector $S_i^{Xy} = (S_i^{yX})^T = X_i^T y_i$,
- scalar $s_i^y = y_i^T y_i$, sample size $n_i$,

for reconstructing the (restricted) LMM likelihood. Specifically, to reconstruct (5) with the above given aggregated data, we plug in (8) to get (3), then plug in (3) to get (4), then plug in (3), (4) and (9) to get (5). Similarly, the additional terms in (6) can also be reconstructed by plugging in (8) and (4).

**Selection of variance components**. We test the significance of random effects of each individual covariate by likelihood ratio test. For simplicity we assume the potential random effects are independent and the random intercept always exists, i.e., $V = diag(\sigma_1^2, ..., \sigma_q^2)$ and $\sigma_1^2 > 0$. For the covariate corresponding to variance component $\sigma_k^2, k \geq 2$, we test

$$H_0 : \sigma_1^2 > 0, \sigma_2^2 = ... = \sigma_q^2 = 0 \ vs \ H_1 : \sigma_1^2 > 0, \sigma_k^2 > 0, \sigma_2^2 = ... = \sigma_{k-1}^2$$
$$= \sigma_{k+1}^2 = ... = \sigma_q^2 = 0.$$

The likelihood ratio test (LRT) statistic given by the likelihood ratio

$$LR = -2\{sup_{H_0} L(\beta, \sigma^2, V) - sup_{H_1} L(\beta, \sigma^2, V)\}, \quad (10)$$

follows a 50:50 mixture of $\chi_0^2$ and $\chi_1^2$ [44,45] under $H_0$. Both the log-likelihoods in (10) can be reconstructed by the communicated aggregated data. Notice that if the potential random effects are not independent, e.g., matrix $V$ admits an unconstrained structure, the distribution of the above test statistics is more complicated and may depends on $V$[44–46].

**Best linear unbiased predictors for the random effects**. Finally, the BLUP[17] of the random effects $u_i$ at the $i^{th}$ site is

$$\hat{u}_i = \hat{\Theta} Z_i^T \Gamma_i(\hat{\Theta})^{-1}(y_i - X_i\hat{\beta}). \quad (11)$$

Conditioning on $X_i$, $\hat{u}_i$ has mean zero and covariance matrix

$$\text{Var}(\hat{u}_i|X_i) = \hat{\Theta} Z_i^T \left[\hat{\sigma}^2 \Gamma_i(\hat{\Theta})^{-1} - \left\{\hat{\sigma}^2 \Gamma_i(\hat{\Theta})^{-1} X_i \left(\sum_{i=1}^{K} X_i^T \Gamma_i(\hat{\Theta})^{-1} X_i\right)^{-1} X_i^T \Gamma_i(\hat{\Theta})^{-1}\right\}\right] Z_i \hat{\Theta}.$$

Since we are more interested in prediction of $u_i$, it is more appropriate to use prediction intervals as below

$$\text{Var}(\hat{u}_i - u_i) = V - \hat{\Theta} Z_i^T \left[\hat{\sigma}^2 \Gamma_i(\hat{\Theta})^{-1} - \left\{\hat{\sigma}^2 \Gamma_i(\hat{\Theta})^{-1} X_i \left(\sum_{i=1}^{K} X_i^T \Gamma_i(\hat{\Theta})^{-1} X_i\right)^{-1} X_i^T \Gamma_i(\hat{\Theta})^{-1}\right\}\right] Z_i \hat{\Theta}.$$

We summarize the analysis with the proposed DLMM algorithm as follows, see Box 1.

**Reporting summary**. Further information on research design is available in the Nature Research Reporting Summary linked to this article.

## Data availability

Part of the data (aggregated data from CCAE, MDCR, STARR, CUIMC, Optum EHR, TRDW, SIDIAP, HIRA COVID) are collected by the OHDSI DistributedLMM protocol (https://github.com/ohdsi-studies/DistributedLMM) and processed by J.R.. Other data (UHG by M.N.I., OneFlorida by Z.C., Optum COVID by Y.Z.) are processed by the corresponding co-authors. All the aggregated data are sent to the first author (C.L.) for final analysis. The EHR/claims data are proprietary and are not publicly accessible due to restricted user agreement. The detailed data description and IRB statements of each data sets are in the Supplementary Notes. For replication purpose, the dataset from the OneFlorida consortium can be obtained by contacting Dr. Jiang Bian (email: bianjiang@ufl.edu) upon the completion of the data usage agreement.

## Code availability

The R code for running DLMM is wrapped in the R (version >= 3.5.0) package "pda" version 1.0–2, available at CRAN (https://CRAN.R-project.org/package=pda) or github (https://github.com/Penncil/pda). A separate documentation using simulated data is at https://github.com/Penncil/DLMM.

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

## Acknowledgements

This work was supported partially through a Patient-Centered Outcomes Research Institute (PCORI) Project Program Award (ME-2019C3-18315, C.L., J.T., R.D., M.E., A.M. and Y.C.). All statements in this report, including its findings and conclusions, are solely those of the authors and do not necessarily represent the views of the Patient-Centered Outcomes Research Institute (PCORI), its Board of Governors or Methodology Committee. Y.C.'s effort was supported in part by National Institutes of Health (1R01LM012607, 1R01AI130460, 1R01AG073435, 1R01LM013519, 1R56AG074604, 1R56AG069880). The research at Ajou University was funded by the Bio Industrial Strategic Technology Development Program (20003883, 20005021, R.W.P.), the Ministry of Trade, Industry & Energy (MOTIE, Korea) and a grant from the Korea Health Technology R&D Project through the Korea Health Industry Development Institute (KHIDI), funded by the Ministry of Health &Welfare, Republic of Korea (grant number: HR16C0001, R.W.P.).

## Author contributions

C.L., and Y.C. devised the project. N.E.S and J.R. coordinated the data collection. C.L. and M.N.I. performed the analysis. C.L., M.N.I., and Y.C. wrote the paper. N.E.S., M.J.S., P.B.R., M.E., R.D., J.T., A.M.A., E.L., J.A.D., R.M.W. and D.A.A. provided guidance on the writing. All other authors provided data for the analysis.

## Competing interests

N.E.S., M.N.I and J.B. are/were full-time employees in Optum Labs and own stock in its parent company, UnitedHealth Group, Inc. J.R. and M.J.S. are full-time employees of Janssen Research and Development. Other authors declare no competing interests.
