## [Peer Review File · Nature Communications]

Reviewers' Comments:

Reviewer #1:

Remarks to the Author:

The manuscript presents a method for distributed estimation of parameters in a linear mixed effects model (LMM). The approach is sound, and the topic extremely timely. There is a clear need for such algorithms, and the proposal here is a clever proposal. The method is limited to LMMs, which is clearly a limitation in practice, but extension to the computationally much more complex case of GLMM is evoked. It is fair to start with this simpler case, and this does not diminish the value of the study.

The method is used to analyze data of COVID-19 patients.

Major comments

1. From a clinical point of view, length of stay (LOS) is relevant in COVID-19, though less than death, need for ventilation support, etc. Moreover, it is also challenging. Depending in the outcome (death or discharge alive), a similar LOS cannot be interpreted similarly. Actually, a shorter LOS with a treatment because patients die more would not be desirable. It is therefore not certain LOS is an ideal outcome to be modelled by linear regression. But suggesting Poisson regression is not much wiser, and the outcome likely calls for much more complex models (see Harhay et al.. Measuring and Analyzing Length of Stay in Critical Care Trials. Med Care. 2019;57:e53-e59).
2. The description of the International COVID-19 hospital LOS study in the results is much too long. This is basically methods, and details such as the table 1 could better fit into the supplementary material. The same for figure 5.
3. Still on that example, I failed to understand the need to see the BLUPs by dataset and discuss the results that way. Random effects assume that datasets can be considered as drawn from a larger population, and model how one varies compared to the average (the common effect, for a given covariate). I agree that the distribution of random effects carries a lot of information but interpreting which dataset had larger or smaller association, without trying to relate this to observable (fixed effects) characteristics does not seem so useful. It would be more informative to compare the spread among datasets to the common effects. It could also be interesting to use two-dimensional bubble plots to study 2-by-2 associations between dataset-specific effects (BLUPs).
4. I assume from the formulas that any formulation of the random effects variance matrix V can be chosen. The test for random effects is however more complex with more general structures of variance, and this could be acknowledged. Moreover, the LRT shown for variance components only tackles an unusual case as soon as more than one covariate would be used: in the general formulation, we could want to test one of the sigma square being null, with no condition on the others (or assuming they are all > 0). Some insights in the mixture and degrees of freedom for this case could be given. Refs 17 and 18 tackle those cases.

Minor comments

1. The order of references is difficult to follow: neither citation order nor alphabetical.
2. It might be due to the journal format, but it seems quite unusual to me to see the model (which is mostly a basic LMM, the value of the paper being the distributed estimation in the results, and details on the fitting procedure in the methods).
3. Just before equation (1), it could be stated that u_i is a vector. Otherwise the reader has to wait until the definition of V to be sure.
4. Some efforts could be made to have all panels of a same figure (e.g. figure 3) having a similar global shape (size of axis fonts, whether axis labels are parallel to the axis or to the reading direction, etc.)
5. "Association" should be preferred over "effect" whenever possible.

Reviewer #2:

Remarks to the Author:

Paper Summary:

This paper proposes a lossless distributed method for training a linear mixed model (LMM) on the

data of multiple sites without sharing individual patient data which is crucial for protecting the privacy of the data subjects. The proposed method works in a one-shot manner, i.e., only one communication round is required to train the linear mixed model. To evaluate the accuracy of their distributed method the authors employ a dataset consisting of medical claims and EHR data from multiple sources across the world and perform a large-scale analysis to predict COVID-19 hospitalization length of stay. Their results demonstrate that the LMM trained in a distributed fashion is identical to that trained in a centralized manner, i.e., when all the data is pulled together, and yield some interesting observations about the factors that affect COVID-19 hospitalization.

Main Comments:

The paper is timely, as sharing medical data among multiple institutions to better understand and fight the COVID-19 pandemic is of tremendous importance. The authors have managed to pull together a big EHR/medical claims dataset from multiple sites across the world and the training of the LMM with their distributed method yields accurate results (compared to a centrally trained model on the data pulled together) and some interesting insights about the factors that affect COVID-19 hospitalization length of stay. However, this reviewer has some concerns about the paper and its proposed method which are discussed below:

The system model is not described clearly and the reader cannot directly grasp the benefit of the one-shot distributed method proposed. How are the multiple sites interconnected to share their data? Is there an entity or a platform that orchestrates the distributed learning process? Is each site training its own model on the shared data? Does every participant get access to the globally trained linear mixed model? Without these details about the workflow of the proposed system and method, medical institutions will not be able to comprehend the merits of joining such a collaboration nor appreciate the benefits of one-shot communication when it comes to collaborative learning with others.

The envisioned threat model is missing from the paper. The authors design a method that aims at protecting the privacy of the data subjects but it is not clear who is the considered adversary. Is it the parties involved in the data sharing process, network eavesdroppers, etc., and what are their expected behaviors (active, passive, etc.)? The proposed method does not require sharing individual patient level data, however, it is well-known that aggregate level data do not protect individuals' privacy from inference attacks (e.g., data reconstruction attacks, membership/attribute inference, etc.) which can result in unexpected information leakage. In fact, the database/computer security communities have established the framework of differential privacy for sharing aggregate level data while limiting the privacy leakage for individuals. Without applying this framework (e.g., see <https://journalprivacyconfidentiality.org/index.php/jpc/article/view/627> for the case of linear mixed models) or other privacy-enhancing techniques while training a joint model (e.g., multi-party computation or homomorphic encryption) the proposed method can *not* be privacy-preserving. Indeed, the fact that the distributed learning method is lossless can be seen as an indication that it does not protect privacy: to achieve privacy, typically a method designer has to "pay" for something else (e.g., a performance or accuracy hit). This reviewer would encourage the authors to clearly define what their privacy goals are, what type of adversaries are envisioned, and formally analyze the privacy achieved by their method. This analysis should be in place, in particular if the authors are heading towards creating a platform that medical institutions will trust and use to share COVID-19 data that are related to their patients.

The proposed method yields an accurate model (compared to a central model trained on all the data pulled together) as well as some interesting insights about the fixed and random effects when it comes to COVID-19 hospitalization showing how LMMs can account for heterogeneous data. However, the scalability of the proposed method is not evaluated: how does the proposed distributed learning method scale with the number of participants, the size of the datasets (samples and features)? Without this information a medical institution would not be in position of deciding whether to join such an initiative. Moreover, the method is not quantitatively compared (performance and accuracy) to other techniques that could be used for the same purpose. For instance, what are the gains of the proposed method compared to meta-analysis techniques?

Similarly, what are the benefits of the method with respect to federated learning (FL) approaches? Recent works show that (variations of) FL can cope very well with heterogeneous (e.g., non-iid) settings and result in models with strong predictive performance. At the same time, FL would allow the training of more complex models (e.g., neural networks) that can potentially offer deeper insights about the use-case at hand compared to a linear mixed model. Such comparisons are essential for convincing the readers about the usefulness and practicality of the proposed method.

More worryingly, the authors seem to be unaware of several recent proposals aiming at secure medical data sharing, see e.g. the ones listed in Table 1 of Froelicher et al.

Truly Privacy-Preserving Federated Analytics for Precision Medicine with Multiparty Homomorphic Encryption

<https://www.biorxiv.org/content/10.1101/2021.02.24.432489v1>

Other Comments:

There is an inconsistency regarding the aggregate values that each site communicates between Figure 2 and the text description on page 23. Please be more precise about the aggregate information required to communicate among the parties.

The proposed method allows for the analysis of site-specific random effects which raises an additional concern for privacy leakage. It is highly possible that such analysis leaks information specific to certain sites (e.g., processes, treatments, etc.) which might be something that an institution would like to avoid. In practice, a good privacy-preserving method would hide this information and reveal only the common fixed effects (or general knowledge) that exist on the data of multiple institutions.

The caption of Figure 6 mentions 13 worldwide data sources. However, the text description (with the UHG broken down into 4 sites) mentions 14 data sources. Please fix.

We thank the editor and two reviewers for their constructive comments and suggestions which resulted in an improved paper. In the revision we have addressed all the reviewers' comments. See below the point-by-point response to the comments, with the response in *Italic* font. All the track changes made to the paper have also been marked in the revised manuscript.

REVIEWER COMMENTS

Reviewer #1 (Remarks to the Author):

The manuscript presents a method for distributed estimation of parameters in a linear mixed effects model (LMM). The approach is sound, and the topic extremely timely. There is a clear need for such algorithms, and the proposal here is a clever proposal. The method is limited to LMMs, which is clearly a limitation in practice, but extension to the computationally much more complex case of GLMM is evoked. It is fair to start with this simpler case, and this does not diminish the value of the study.

The method is used to analyze data of COVID-19 patients.

Major comments

R1Q1. From a clinical point of view, length of stay (LOS) is relevant in COVID-19, though less than death, need for ventilation support, etc. Moreover, it is also challenging. Depending in the outcome (death or discharge alive), a similar LOS cannot be interpreted similarly. Actually, a shorter LOS with a treatment because patients die more would not be desirable. It is therefore not certain LOS is an ideal outcome to be modelled by linear regression. But suggesting Poisson regression is not much wiser, and the outcome likely calls for much more complex models (see Harhay et al.. Measuring and Analyzing Length of Stay in Critical Care Trials. Med Care. 2019;57:e53-e59¹).

Response: *We thank the reviewer for this important comment. In this paper we use COVID-19 LOS study as an illustrating example for the proposed DLMM algorithm. It's true that the interpretation of LOS depends on the outcome, e.g. a patient who eventually died may have a shorter LOS. To address this problem, and because a majority of the COVID-19 patients survived, we restrict the study cohort to the patients who are discharged alive. See the revised Supplementary Materials page 6 lines 6-12:*

"In the OHDSI study patients were included into the cohort if they had an inpatient visit between January 2020 and September 2020 satisfying

- *age 18 years or older*
- *A COVID-19 diagnosis or positive test recorded up to 21 days prior to the visit or during the visit*
- *Been active in the database for 6 months or more prior to the inpatient visit*

- Did not have a discharge status of "expired" prior to September 30, 2020."

As a result, the data of 9 sites, i.e. the sizes of sites except UHG and HIRA COVID have been changed, see Tables S3 (previous Table 1) and S4 for details, the total sample size is now 120,609 (was 119,235). Notice that this slight change of sample does not produce qualitatively different conclusion about the study. For example, the estimates of the fixed effects shown in previous and revised Figure 6 have small differences.

Previous Fig 6

Revised Figure 6

On the other hand, the generalized linear mixed model (GLMM) may be used for more general types of outcomes, e.g. logistic regression for binary outcome (need for ventilation support, death, etc), or Poisson regression for treating LOS as a count outcome. A distributed algorithm for GLMM is thus desired and considered as one of our future directions. See Discussion, the end of page 12:

“We treated LOS as a continuous outcome mainly for the purpose of illustrating the proposed distributed algorithm. It would be more reasonable to model LOS as a count outcome via the generalized linear mixed model (GLMM) framework, e.g., Poisson regression with mixed effects, to account for the between-site heterogeneity. Other important outcomes, such as mortality, can also be modeled within the GLMM framework. ... Distributed algorithms for GLMM estimation are currently under investigation and will be reported in the future.”

R1Q2. The description of the International COVID-19 hospital LOS study in the results is much too long. This is basically methods, and details such as the table 1 could better fit into the supplementary material. The same for figure 5.

Response: We thank the reviewer for the suggestion. We now move the data description (Table 1 and the itemized description) to the Supplementary Materials. We keep Figure 5 in the manuscript as a brief introduction of the collaborative databases.

R1Q3. Still on that example, I failed to understand the need to see the BLUPs by dataset and discuss the results that way. Random effects assume that datasets can be considered as drawn

from a larger population, and model how one varies compared to the average (the common effect, for a given covariate). I agree that the distribution of random effects carries a lot of information but interpreting which dataset had larger or smaller association, without trying to relate this to observable (fixed effects) characteristics does not seem so useful. It would be more informative to compare the spread among datasets to the common effects. It could also be interesting to use two-dimensional bubble plots to study 2-by-2 associations between dataset-specific effects (BLUPs).

Response: We thank the reviewer for suggesting a better presentation of the random effects estimates. We agree that relating the BLUPs to the estimated fixed effects is a better approach. Following your advice, we now present $\beta + u_i$ and its 95% confidence interval, rather than the confidence interval for u_i only in Figure 7.

R1Q4. I assume from the formulas that any formulation of the random effects variance matrix V can be chosen. The test for random effects is however more complex with more general structures of variance, and this could be acknowledged. Moreover, the LRT shown for variance components only tackles an unusual case as soon as more than one covariate would be used: in the general formulation, we could want to test one of the sigma square being null, with no condition on the others (or assuming they are all > 0). Some insights in the mixture and degrees of freedom for this case could be given. Refs 17 and 18 tackle those cases.

Response: We thank the reviewer for raising this specific question regarding random effect testing. It's true that any formulation of the random effects variance matrix V can be chosen, and the test for random effects is more complex with a general unconstrained structure of V . For example, the distribution of the likelihood ratio test statistic is a 50-50 mixture of a chi-square-0 and a chi-square-1 distributions when V is diagonal, but may be more complicated when V is unconstrained. Since this is not the focus of our paper, we use the simple case to illustrate our method where the random effects are independent, i.e. V is diagonal. Also, a LMM with an unconstrained V may involve many more parameters and is difficult to estimate. For example, in our study we have 16 covariates (including intercept) and it makes less sense to estimate a 16-by-16 V matrix. In our software package, we allow users specify either diagonal or unstructured V matrix when they fit DLMM, if the number of covariates is appropriate.

We now use sequential selection of the significant random effects, i.e. forward select the random effects starting from a model with random intercept only (assume random effects are independent). This results in 15 significant random effects, see Table S5. Compared to previous univariate selection which results in all 16 covariates being significant, the forward selection is a more reasonable selection procedure and obtains a more parsimonious model.

See the end of page 14:

“... Notice that if the potential random effects are not independent, e.g. matrix V admits an unconstrained structure, the distribution of the above test statistics is more complicated and may depend on V .”

Minor comments

1. The order of references is difficult to follow: neither citation order nor alphabetical.

Response: *we now arrange the references in citation order.*

2. It might be due to the journal format, but it seems quite unusual to me to see the model (which is mostly a basic LMM, the value of the paper being the distributed estimation) in the results, and details on the fitting procedure in the methods.

Response: *Nature Communication requires the Results section after Introduction, and Methods section in the end of the manuscript. In order to introduce the notations that used for the proposed DLMM algorithm (e.g. aggregated data $S_i^X = X_i^T X_i$, $S_i^{Xy} = X_i^T y_i$, $s_i^y = y_i^T y_i$ and sample size n_i from the i^{th} site etc), we thus start from introducing the basic LMM setting.*

3. Just before equation (1), it could be stated that u_i is a vector. Otherwise the reader has to wait until the definition of V to be sure.

Response: *We now add “ u_i is the q -dimensional random effect” before equation (1).*

4. Some efforts could be made to have all panels of a same figure (e.g. Figure 3) having a similar global shape (size of axis fonts, whether axis labels are parallel to the axis or to the reading direction, etc.)

Response: *The figures (e.g. Figure 3) are now modified to have a consistent presentation.*

5. “Association” should be preferred over “effect” whenever possible.

Response: *We now change “effect” to “association” when interpreting the estimation results.*

Reviewer #2 (Remarks to the Author):

Paper Summary:

This paper proposes a lossless distributed method for training a linear mixed model (LMM) on the data of multiple sites without sharing individual patient data which is crucial for protecting the privacy of the data subjects. The proposed method works in a one-shot manner, i.e., only one communication round is required to train the linear mixed model. To evaluate the accuracy of their distributed method the authors employ a dataset consisting of medical claims and EHR data from multiple sources across the world and perform a large-scale analysis to predict COVID-19 hospitalization length of stay. Their results demonstrate that the LMM trained in a distributed fashion is identical to that trained in a centralized manner, i.e., when all the data is pulled together, and yield some interesting observations about the factors that affect COVID-19 hospitalization.

Main Comments:

The paper is timely, as sharing medical data among multiple institutions to better understand and fight the COVID-19 pandemic is of tremendous importance. The authors have managed to pull together a big EHR/medical claims dataset from multiple sites across the world and the training of the LMM with their distributed method yields accurate results (compared to a centrally trained model on the data pulled together) and some interesting insights about the factors that affect COVID-19 hospitalization length of stay.

Response: *Thank you for your positive review of this work. Hereafter, we are addressing the concerns from you and have made corresponding the revisions in our revised manuscript.*

R2Q1: However, this reviewer has some concerns about the paper and its proposed method which are discussed below:

The system model is not described clearly and the reader cannot directly grasp the benefit of the one-shot distributed method proposed. How are the multiple sites interconnected to share their data? Is there an entity or a platform that orchestrates the distributed learning process? Is each site training its own model on the shared data? Does every participant get access to the globally trained linear mixed model? Without these details about the workflow of the proposed system and method, medical institutions will not be able to comprehend the merits of joining such a collaboration nor appreciate the benefits of one-shot communication when it comes to collaborative learning with others.

Response: *The reviewer asked a set of great questions. We address them one-by-one in what follows.*

R2Q1a: The system model is not described clearly and the reader cannot directly grasp the benefit of the one-shot distributed method proposed. How are the multiple sites interconnected to share their data?

Response: *Our work with the emphasis on the one-shot distributed algorithm was largely motivated from our participation and collaborations within the Observational Health Data Sciences and Informatics (OHDSI) community. However, the one-shot distributed method can be applied in networks of many other secure healthcare databases^{12,13,14}. In these databases, the computational application programming interface (API) facilitating iterative data communication is usually not feasible, hence the usage of traditional federated learning is limited.*

We elaborate more about the OHDSI setting as an example. The OHDSI community is an international, interdisciplinary collaborative consortium whose network spans more than 600 million patients and more than 2700 researchers. Within this OHDSI community, each data owner keeps their own data at their institute, and participates a collaborative project by sharing their aggregated data through SSH File Transfer Protocol (sftp) or secure email communications. From our experiences of collaborations within OHDSI, it is critical for the data owners to be able to manually review the aggregated data to be shared. In this case, the automated data transfer (e.g., API portal at data server) is less preferred. This is indeed a key difference with many of the existing federated learning algorithms, with automated iterative communications across data owners.

Over the past years, collaboration within the OHDSI community has produced large-scale high-quality evidence via this research network; see for example, the comparative effectiveness studies of hypertension interventions², a study on the effect of RAS inhibitors on COVID-19³, or the safety of hydroxychloroquine⁴. The new evidence generated from the multi-site studies has greatly advanced the methodology and knowledge on drug efficacy, drug safety, health policy, and other fields, using real-world settings.

Since not everyone is familiar with OHDSI, we agree that it is helpful to clarifying the motivation for one-shot algorithms and the actual workflow of data sharing, which was missing in our manuscript. We now added a detailed description and a diagram in the Supplementary Materials, see page 3:

- 1. Project Initiation – Protocol Development: the project leaders initiate a project and develop an analysis protocol (attached at the end of this response letter, and also submitted as a research supplementary material in our submission of revision). In addition to the analysis protocol, the project leaders need to prepare and test computing programs (e.g., R programs) that are ready for the participating sites to run at their local site (usually data in OMOP CDM) with results prepared in the right format;*
- 2. Recruitment of participating sites: the project leaders post the research project to the OHDSI forum (currently in Microsoft Team) with a deadline of 2-3 weeks for the participants to comment, ask questions, and join this project;*
- 3. Communication: the project leaders create an email list or SSH File Transfer Protocol (sftp) file sharing platform with the contact persons from all participating sites, distribute the prepared R program, and set a deadline for returning the results (of aggregated data) (usually 4 weeks). During these four weeks, the participating sites can ask questions that they encounter during the application of the algorithms. At the end of four weeks, the*

participating sites share all the requested results (i.e., aggregated data specified by the algorithm);

4. Aggregation of multi-site results and submitting the final results: the project leaders conduct the final analyses by aggregating the results from all participating sites, and share the results to all participants in a written manuscript for comments before submission.

R2Q1b: Is there an entity or a platform that orchestrates the distributed learning process?

Response: Currently, the study was initiated on the OHDSI forum and the results were shared via email communications. See the workflow in the answer to **R2Q1a**. Such process is less automated, but the data owner has full control on the data to be shared. As added to the Discussion Section, we are currently building a web platform for secure data sharing coupled with our DLMM algorithm to facilitate multi-center collaboration and allow all sites join the collaboration and contribute their aggregated data conveniently.

R2Q1c: Is each site training its own model on the shared data? Does every participant get access to the globally trained linear mixed model?

Response: For clarification, instead of each site training its own model, the proposed DLMM algorithm aims for training **a unified model** while accounting for between-site heterogeneity. Specifically, each site shares the aggregated data to all participating sites, so that any site can build the globally trained linear mixed model by themselves. Such design enables the true decentralization of the algorithm and transparency/reproducibility of the analysis results.

R2Q1d: Without these details about the workflow of the proposed system and method, medical institutions will not be able to comprehend the merits of joining such a collaboration nor appreciate the benefits of one-shot communication when it comes to collaborative learning with others.

Response: We have added a new diagram on the workflow for the development of the algorithm. We also provide a step-by-step description of the workflow. Hopefully this will improve the clarity of the proposed DLMM algorithm. See the answer to **R2Q1a**.

R2Q2: The envisioned threat model is missing from the paper. The authors design a method that aims at protecting the privacy of the data subjects but it is not clear who is the considered adversary. Is it the parties involved in the data sharing process, network eavesdroppers, etc., and what are their expected behaviors (active, passive, etc.)? The proposed method does not require sharing individual patient level data, however, it is well-known that aggregate level data do not protect individuals' privacy from inference attacks (e.g., data reconstruction attacks, membership/attribute inference, etc.) which can result in unexpected information leakage. In fact, the database/computer security communities have established the framework of differential privacy for sharing aggregate level data while limiting the privacy leakage for individuals. Without applying this framework (e.g., see <https://journalprivacyconfidentiality.org/index.php/jpc/article/view/627> for the case of linear mixed models) or other privacy-enhancing techniques while training a joint model (e.g., multi-party computation or homomorphic encryption) the proposed method can *not* be privacy-preserving. Indeed, the fact that the distributed learning method is lossless can be seen as an indication that it does not protect privacy: to achieve privacy, typically a method designer has to “pay” for something else (e.g., a performance or accuracy hit). This reviewer would encourage the authors to clearly define what their privacy goals are, what type of adversaries are envisioned, and formally analyze the privacy achieved by their method. This analysis should be in place, in particular if the authors are heading towards creating a platform that medical institutions will trust and use to share COVID-19 data that are related to their patients.

Response: *We thank the reviewer for raising these important questions. We are addressing them one-by-one below.*

R2Q2a: The authors design a method that aims at protecting the privacy of the data subjects but it is not clear who is the considered adversary. Is it the parties involved in the data sharing process, network eavesdroppers, etc., and what are their expected behaviors (active, passive, etc.)?

Response: *As we clarified in our response to the R2Q1, this algorithm was motivated from OHDSI – an international collaborative research community, who has been sharing aggregated data over the last 10 years for collaborative projects. The participants joined OHDSI as volunteers and most of them are active research participants. On the other hand, when we design our algorithm, we have tried our best to minimize the workload/request for the participating sites. Indeed, the contact person at the participating site only needs to run a specific program and share the results from running that program. There was only one-round of communication involved. Per your advice, we have added a diagram to explain the workflow; see Figure S4 in the revised Supplementary Materials.*

Possible adversary is from collaborators that participate in the study through passive attack, i.e. re-identify patients' partial protected health information (PHI, e.g. i.e. COVID-19 status) by linking the released aggregated data with some external databases. This is possible when some characteristics are extremely rare (e.g. only one patient in a certain site has cancer). To further avoid this possibility in future collaboration, we will require the collaborative sites to have an adequate number of patients (e.g. above 500), and review the aggregated data to avoid sparse cells (e.g. no cell count less than 5) before submitting it in case some characteristics are extremely rare.

R2Q2b: The proposed method does not require sharing individual patient level data, however, it is well-known that aggregate level data do not protect individuals' privacy from inference attacks (e.g., data reconstruction attacks, membership/attribute inference, etc.) which can result in unexpected information leakage.

Response: The reviewer's comment reminds us to further discuss the privacy-protection of our aggregated data release mechanism. See the Discussion section page 12, lines 1-16:

"Our DLMM algorithm is considered privacy-preserving as it only requires one-shot communication of aggregated data from collaborative sites, and the aggregated data are only shared within collaborators who participate in the study. However, our aggregated data release mechanism has not been rigorously studied to meet privacy-preserving criteria such as k -anonymity or differential privacy²⁷⁻²⁹. Specifically, the k -anonymity property protects against the risk of re-identification²⁷, which arises from linking potential quasi-identifiers (i.e. combinations of patient's characteristics in our study) to external sources²⁷. In DLMM aggregated data, if all the cell counts are not sparse, the algorithm can potentially meet the k -anonymity requirement. In future collaborations using DLMM, we suggest data contributors review the aggregated data to avoid sparse cells (e.g. no cell count less than 5) before sending them to other sites. We will quantify the risk of privacy leaking more rigorously, and enhance our DLMM algorithm via techniques such as differential privacy and multiparty homomorphic encryption³⁰ in the future. On the other hand, when reporting the estimated BLUPs for site-specific random effects, caution must be exercised if there is sensitive information specific to sites that could be inferred from the result. We thus suggest the estimated BLUPs not be

disclosed if the covariates are sensitive characteristics and write this in the protocol when initiating a collaborative project using the DLMM algorithm in the future.”

On the other hand, the aggregated data (e.g., counts, averages, standard deviations, and Kaplan-Meier curves as estimated survival functions) are commonly reported in medical literature (e.g., the summary Table of patients’ characteristics in a cohort in many medical papers). Earlier algorithms in distributed regression, such as the iterative distributed logistic regression (GLORE) and iterative distributed Cox regression (WebDISCO), require sharing of weighted matrices (e.g., $X^T W X$ and $X^T W Y$) iteratively. They were considered as privacy-preserving algorithms in biomedical informatics community, assuming there are large enough number of patients at each site. Compared to these existing algorithms, the proposed DLMM algorithm has less privacy concerns, as it only requires one round of communication. We believe that the DLMM algorithm would work in the OHDSI and other clinical research network settings.

In our international study of COVID-19 LOS, to further illustrate what summary statistics were actually communicated, we present below an example of the aggregated data from a synthetic data of 2,000 patients, with the first several patients’ data in Figure R1. All the covariates are binary variables and the outcome LOS is integer days.

	age_65_80	age_80_	CCI_2_5	CCI_5_	gender_male	adm_Q2	adm_Q3	cancer	copd	hypertension	hyperlipidemia	kidney_disease	obesity	heart_disease	diabetes	LOS
1	0	1	0	0	0	0	1	0	1	1	1	1	0	1	0	8
2	0	1	0	0	1	1	0	0	1	0	0	0	1	1	0	10
3	0	1	1	0	1	0	0	0	0	0	0	0	0	1	1	8
4	0	0	1	0	0	1	0	0	1	0	0	0	1	0	0	9
5	0	0	1	0	1	0	1	1	0	0	0	0	0	0	0	5
6	1	0	0	0	1	0	1	0	0	1	0	0	1	1	0	5
7	1	0	1	0	1	0	1	0	0	1	0	0	0	0	0	4
8	0	0	0	1	0	1	0	0	0	0	1	1	1	0	0	6
9	1	0	0	0	0	1	0	0	0	1	0	0	0	0	0	10
10	0	1	0	1	0	1	0	0	0	0	0	0	0	0	0	8

Showing 1 to 10 of 2,000 entries, 16 total columns

Figure R1. Example (synthetic) patient-level data.

For our DLMM algorithm, the shared aggregated data from a participating site are presented in Figure R2. In fact, in our real-world study, all the numbers in the aggregated data were integers representing counts. For example, the (2,2) cell 958 is the count of patients with age between 65 and 80, and the (2,6) cell 466 is the count of male patients with age between 65 and 80.

	Intercept	age_65_80	age_80_	CCI_2_5	CCI_5_	gender_male	adm_Q2	adm_Q3	cancer	copd	hypertension	hyperlipidemia	kidney_disease	obesity	heart_disease	diabetes	LOS
Intercept	2000	958	531	1007	488	981	974	503	606	597	632	576	559	595	634	598	15983
age_65_80	958	958	0	476	241	466	479	232	280	285	311	270	259	278	295	292	7579
age_80_	531	0	531	286	120	251	250	143	174	161	168	155	143	166	176	156	4246
CCI_2_5	1007	476	286	1007	0	511	478	253	305	295	307	302	285	299	332	289	8009
CCI_5_	488	241	120	0	488	230	230	129	146	146	152	147	129	155	150	152	3910
gender_male	981	466	251	511	230	981	481	241	280	287	316	280	277	303	277	296	7817
adm_Q2	974	479	250	478	230	481	974	0	287	281	304	299	276	284	300	294	7747
adm_Q3	503	232	143	253	129	241	0	503	160	151	153	131	139	153	179	155	3977
cancer	606	280	174	305	146	280	287	160	606	181	203	161	172	178	205	179	4846
copd	597	285	161	295	146	287	281	151	181	597	194	173	175	173	190	175	4722
hypertension	632	311	168	307	152	316	304	153	203	194	632	181	188	183	205	204	5086
hyperlipidemia	576	270	155	302	147	280	299	131	161	173	181	576	148	190	168	179	4671
kidney_disease	559	259	143	285	129	277	276	139	172	175	188	148	559	158	181	176	4507
obesity	595	278	166	299	155	303	284	153	178	173	183	190	158	595	195	181	4714
heart_disease	634	295	176	332	150	277	300	179	205	190	205	168	181	195	634	191	5106
diabetes	598	292	156	289	152	296	294	155	179	175	204	179	176	181	191	598	4818
LOS	15983	7579	4246	8009	3910	7817	7747	3977	4846	4722	5086	4671	4507	4714	5106	4818	141125

Figure R2. Example aggregated data.

R2Q2c: In fact, the database/computer security communities have established the framework of differential privacy for sharing aggregate level data while limiting the privacy leakage for individuals. Without applying this framework (e.g., see <https://journalprivacyconfidentiality.org/index.php/jpc/article/view/627> for the case of linear mixed models¹⁰) or other privacy-enhancing techniques while training a joint model (e.g., multi-party computation or homomorphic encryption⁹) the proposed method can **not** be privacy-preserving.

Response: *We are well aware of the relevant privacy literature. We agree that incorporation with differential privacy or homomorphic encryption can add more rigorous justification of the privacy-protection property of our algorithms. This will be an interesting area for our future investigation. In the revised manuscript, we have added a discussion along this line. See the Discussion section, page 12, lines 3-12:*

“...However, our aggregated data releasing mechanism has not been rigorously studied to meet privacy-preserving criteria such as k-anonymity or differential privacy²⁷⁻²⁹. ... In future collaborations using DLMM, we suggest data contributors review the aggregated data to avoid sparse cells (e.g. no cell count less than 5) before sending them to other sites. We will quantify the risk of privacy leaking more rigorously, and enhance our DLMM algorithm via techniques such as differential privacy and multiparty homomorphic encryption³⁰ in the future.”

R2Q2d: Indeed, the fact that the distributed learning method is lossless can be seen as an indication that it does not protect privacy: to achieve privacy, typically a method designer has to “pay” for something else (e.g., a performance or accuracy hit).

Response: *We agree that in normal circumstances, an algorithm being lossless has to pay for something else (such as privacy, or communication-efficiency). However, there are rare exceptions algorithms can be both lossless and privacy-protected. One such example is the lossless distributed algorithm for linear regression (Chen et al., 2006¹¹), and our proposed algorithm inherited such unique property from the distributed algorithm for linear regression by Chen et al.¹¹, yet being able to account for between-study heterogeneity.*

We thank the reviewer for making this important point (as it may be counterintuitive to have both lossless and privacy-protection). In our revised manuscript, we have made the clarification that due to the properties of linear mixed effect models, our algorithm inherited a unique property as in distributed linear regression for being both lossless and privacy-preserving at the same time.

In the revision, we have added the following on the page 4 of the manuscript:

“We note that, generally, an algorithm being lossless has to sacrifice certain properties of the algorithm, such as the privacy protection or communication-efficiency. However, there are rare exceptions algorithms can be both lossless and privacy-protected. One such example is the lossless distributed algorithm for linear regression, being lossless, communication-efficient and

privacy-protected. Our proposed algorithm inherits such unique property of the distributed linear regression for being both lossless and privacy-preserving, yet being able to account for between-study heterogeneity.”

Thanks again to the reviewer for raising this point for clarifications.

R2Q3: The proposed method yields an accurate model (compared to a central model trained on all the data pulled together) as well as some interesting insights about the fixed and random effects when it comes to COVID-19 hospitalization showing how LMMs can account for heterogeneous data. However, the scalability of the proposed method is not evaluated: how does the proposed distributed learning method scale with the number of participants, the size of the datasets (samples and features)? Without this information a medical institution would not be in position of deciding whether to join such an initiative. Moreover, the method is not quantitatively compared (performance and accuracy) to other techniques that could be used for the same purpose. For instance, what are the gains of the proposed method compared to meta-analysis techniques? Similarly, what are the benefits of the method with respect to federated learning (FL) approaches? Recent works show that (variations of) FL can cope very well with heterogeneous (e.g., non-iid) settings and result in models with strong predictive performance. At the same time, FL would allow the training of more complex models (e.g., neural networks) that can potentially offer deeper insights about the use-case at hand compared to a linear mixed model. Such comparisons are essential for convincing the readers about the usefulness and practicality of the proposed method.

Response: *Here we address these questions one-by-one.*

R2Q3a: However, the scalability of the proposed method is not evaluated: how does the proposed distributed learning method scale with the number of participants, the size of the datasets (samples and features)? Without this information a medical institution would not be in position of deciding whether to join such an initiative.

Response: *For the scalability, the proposed DLMM algorithm scales to large number of participating sites and the sample size of the datasets, as the required aggregated data are only the $p \times p$ matrix and does not involve the sample size n_i . This is in fact another advantage of the DLMM algorithm, as LMM estimation via this algorithm depends on the IPD only through the aggregated data, and thus avoids involving the IPD computation in a central model. However, we have not extended the current model to high-dimensional features (i.e., large p). This would be an interesting topic for future investigation. The current DLMM algorithm can handle a moderate number of features ($p < n$), given n is relatively large (hundreds of patients per site). Such algorithm is sufficient for many of epidemiological models for association analyses.*

In the revised manuscript, we have added the following to the discussion, see page 12, lines 2-7:

“Regarding scalability, our DLMM algorithm has great scalability in terms of large number of sites and large number of patients per site. However, in the presence of high dimensional features (i.e., large p), the current algorithm will require sharing of $p \times p$ dimensional matrices, which may be challenging. Extension of DLMM algorithm to improve the scalability on large number of features remains an important area for future research.”

R2Q3b: Moreover, the method is not quantitatively compared (performance and accuracy) to other techniques that could be used for the same purpose. For instance, what are the gains of the proposed method compared to meta-analysis techniques?

Response: *The proposed method focuses on a presumption that linear mixed effect model (LMM) is appropriate for an association study on impacts of risk factors of a continuous outcome. Under this context, we showed that the DLMM algorithm is communication-efficient and lossless, both mathematically and empirically. There are certainly other methods that can be used. For example, meta-analysis that averages the individual estimates from each site is a simple way to integrate data from multiple sites. Comparing to meta-analysis, a LMM model benefits site-specific prediction due to the shrinkage to the common fixed effects. We believe that the DLMM algorithm achieves a good balance between handling heterogeneity, preserving privacy, accuracy (lossless) and communication efficiency (one round of transferring aggregate data), and comparing to other methods may not be needed.*

R2Q3c: Similarly, what are the benefits of the method with respect to federated learning (FL) approaches? Recent works show that (variations of) FL can cope very well with heterogeneous (e.g., non-iid) settings and result in models with strong predictive performance. At the same time, FL would allow the training of more complex models (e.g., neural networks) that can potentially offer deeper insights about the use-case at hand compared to a linear mixed model. Such comparisons are essential for convincing the readers about the usefulness and practicality of the proposed method.

Response: *We generally agree that the FL models are flexible and useful in many settings. We also consider the proposed method as one variation of FL based on a specific statistical model, i.e. LMM. LMMs are commonly used for its flexibility and interpretability, especially in epidemiological studies (including association analyses). Traditional FL models have a focus on prediction, whilst the LMM model in our analyses of LOS outcome was focusing on quantifying associations of risk factors, which is commonly used in biomedical researches.*

See Discussion, page 13 lines 7-13:

“Lastly, federated learning methods have gained a great deal of attention in many clinical settings in recent years. We also consider the proposed method as one variation of federated learning based on a specific statistical model, i.e. LMM. The LMM holds the promise of flexibility and interpretability of regression coefficients, which are particularly suitable for epidemiological studies. Traditional federated learning models have a focus on prediction, whilst the LMM model in our analyses of LOS outcome focused on quantifying associations of risk factors, which is commonly used in biomedical researches. In the future, we plan to investigate

communication-efficient federated learning algorithms in distributed research network settings, which is a much-needed area for new methods.”

R2Q4: More worryingly, the authors seem to be unaware of several recent proposals aiming at secure medical data sharing, see e.g. the ones listed in Table 1 of Froelicher et al. Truly Privacy-Preserving Federated Analytics for Precision Medicine with Multiparty Homomorphic Encryption⁹

<https://www.biorxiv.org/content/10.1101/2021.02.24.432489v1>

Response: *We thank the reviewer for suggesting us this state-of-the-art reference article. The Multiparty Homomorphic Encryption is truly a useful technique for privacy-preserving of the site-specific data. This technique could potentially be combined with DLMM and further protect the aggregated data from privacy leaking when some covariates are extremely rare. See Discussion, page 12 lines 10-12:*

“... We will quantify the risk of privacy leaking more rigorously, and enhance our DLMM algorithm via techniques such as differential privacy and multiparty homomorphic encryption³⁰ in the future.”

Other Comments:

1. There is an inconsistency regarding the aggregate values that each site communicates between Figure 2 and the text description on page 23. Please be more precise about the aggregate information required to communicate among the parties.

Response: *We thank the reviewer to point out this inconsistency. To avoid notation confusion, we now assume the random effects covariates z_{ij} is part or all of x_{ij} , and thus in the Methods section, the proposed DLMM algorithm requires the i^{th} site to communicate*

- $p \times p$ matrix $S_i^X = X_i^T X_i$,
- $p - \text{dim}$ vector $S_i^{Xy} = (S_i^{yX})^T = X_i^T y_i$,
- scalar $S_i^y = y_i^T y_i$, sample size n_i .

2. The proposed method allows for the analysis of site-specific random effects which raises an additional concern for privacy leakage. It is highly possible that such analysis leaks information specific to certain sites (e.g., processes, treatments, etc.) which might be something that an institution would like to avoid. In practice, a good privacy-preserving method would hide this information and reveal only the common fixed effects (or general knowledge) that exist on the data of multiple institutions.

Response: *We thank the reviewer for reminding us this privacy issue. In this study, site specific information could be inferred from the estimated BLUPs for site-specific random effects (e.g.*

HIRA COVID, Optum EHR and SIDIAP have estimated BLUPs of effect of Q3 admission being 0, meaning that they have no patients admitted during Q3), which is non-sensitive and is already disclosed in the patient characteristic table (Table S4). It's generally true that estimation of BLUPs could leak sensitive information specific to certain sites (e.g., processes, treatments, etc). We thus suggest the estimated BLUPs not be disclosed if the covariates are sensitive characteristics. We will write this in the protocol when initiating collaborative project using the DLMM algorithm in the future. See page 12, lines 12-16:

“On the other hand, when reporting the estimated BLUPs for site-specific random effects, caution must be exercised if there is sensitive information specific to sites that could be inferred from the result. We thus suggest the estimated BLUPs not be disclosed if the covariates are sensitive characteristics and write this in the protocol when initiating a collaborative project using the DLMM algorithm in the future.”

3. The caption of Figure 6 mentions 13 worldwide data sources. However, the text description (with the UHG broken down into 4 sites) mentions 14 data sources. Please fix.

Response: *It's now fixed as “14 sites” to be consistent with the text.*

References

1. Harhay MO, Ratcliffe SJ, Small DS, Suttner LH, Crowther MJ, Halpern SD. Measuring and analyzing length of stay in critical care trials. *Med Care*. 2019;57(9):e53-e53.
2. Suchard MA, Schuemie MJ, Krumholz HM, et al. Comprehensive comparative effectiveness and safety of first-line antihypertensive drug classes: a systematic, multinational, large-scale analysis. *Lancet*. 2019;394(10211):1816-1826.
3. Morales DR, Conover MM, You SC, et al. Renin–angiotensin system blockers and susceptibility to COVID-19: an international, open science, cohort analysis. *Lancet Digit Heal*. 2021;3(2):e98-e114.
4. Lane JCE, Weaver J, Kostka K, et al. Risk of hydroxychloroquine alone and in combination with azithromycin in the treatment of rheumatoid arthritis: a multinational, retrospective study. *Lancet Rheumatol*. 2020;2(11):e698-e711.
5. Sweeney L. k-anonymity: A model for protecting privacy. *Int J Uncertainty, Fuzziness Knowledge-Based Syst*. 2002;10(05):557-570.
6. D’Acquisto G, Domingo-Ferrer J, Kikiras P, Torra V, de Montjoye Y-A, Bourka A. Privacy by design in big data: an overview of privacy enhancing technologies in the era of big data analytics. *arXiv Prepr arXiv151206000*. Published online 2015.
7. Dwork C, McSherry F, Nissim K, Smith A. Calibrating Noise to Sensitivity in Private Data Analysis. *J Priv Confidentiality*. 2017;7(3):17-51. doi:10.29012/jpc.v7i3.405
8. Wasserman L, Zhou S. A Statistical Framework for Differential Privacy. *J Am Stat Assoc*. 2010;105(489):375-389. doi:10.1198/jasa.2009.tm08651
9. Froelicher D, Troncoso-Pastoriza JR, Raisaro JL, et al. Truly Privacy-Preserving Federated Analytics for Precision Medicine with Multiparty Homomorphic Encryption. *bioRxiv*. Published online 2021.
10. Abowd JM, Schneider MJ, Vilhuber L. Differential privacy applications to Bayesian and linear mixed model estimation. *J Priv Confidentiality*. 2013;5(1).
11. Chen Y, Dong G, Han J, Pei J, Wah BW, Wang J. Regression cubes with lossless compression and aggregation. *IEEE Trans Knowl Data Eng*. 2006;18(12):1585-1599.
12. The Department of Veterans Affairs Open Data Portal, <https://www.data.va.gov/>
13. The All of Us Research Program, <https://allofus.nih.gov/>
14. Brat, G.A., Weber, G.M., Gehlenborg, N. et al. International electronic health record-derived COVID-19 clinical course profiles: the 4CE consortium. *npj Digit. Med*. 3, 109 (2020). <https://doi.org/10.1038/s41746-020-00308-0>

Reviewers' Comments:

Reviewer #1:

Remarks to the Author:

I thank the authors for their answers and revisions. There are two points that I could have brought before (sorry for that).

1. The lossless property may be considered as important. One alternative approach that can be used when no individual patients data (IPD) can be shared is to fit the models on each dataset and then meta-analyze the results. Methods have been studied when IPD are available for one center (or study), but not the others (Debray et al. *Statist Med.* 2012;31:2697-2712, Debray et al. *Statist Med.* 2014;33:2341-2362) or using only aggregated information (Sheng et al. *Statist Med.* 2014;33:2567-2576). The focus in those articles was more on logistic or Cox regression than on linear regression, but I wondered how much loss there would be compared to the proposed DLMM approach. I'm not asking for extensive additional analyses but perhaps some discussion could help.

2. Another issue that could be considered when building DLMMs with interactions between variables is the potential for aggregation bias. This has been raised for testing interactions in meta-analyses of randomized trials using IPDs, including linear models, and some solutions have been proposed (Riley et al. *Statist Med.* 2020;39:2115-2137). Again, there is no need to implement that, but the issue could be brought to the readers as a caveat in the discussion.

Reviewer #2:

Remarks to the Author:

- System/Threat Model: The authors seem to have clarified this issue, and motivate it via the OHDSI network example (which they are part of).

- Model Access: Each party gets the resulting global model in clear. While this allows fairness/reproducibility etc., we know that this model access (white-box) is prone to various privacy attacks.

- Privacy Mechanism: Still, besides simple (cleartext) aggregation, the authors have not incorporated any protection mechanism, e.g., encryption or differential privacy: We mentioned that aggregates leak information about individuals and their response to our criticism is "we might do this in the future" or "we will inspect the aggregates before sharing them" (this can not work without any rigorous analysis). Overall, they push the whole criticism to future work as per their response:

"However, our aggregated data releasing mechanism has not been rigorously studied to meet privacy-preserving criteria such as k-anonymity or differential privacy 27–29. ... In future collaborations using DLMM, we suggest data contributors review the aggregated data to avoid sparse cells (e.g. no cell count less than 5) before sending them to other sites. We will quantify the risk of privacy leaking more rigorously, and enhance our DLMM algorithm via techniques such as differential privacy and multiparty homomorphic encryption³⁰ in the future."

- Scalability Analysis: The authors did not do a scalability analysis with respect to the pxp matrix that is being shared by each party. Again, they leave this as future work:

"Regarding scalability, our DLMM algorithm has great scalability in terms of large number of sites and large number of patients per site. However, in the presence of high dimensional features (i.e., large p), the current algorithm will require sharing of high dimensional matrices, which may be challenging. Extension of DLMM algorithm to improve the scalability on large number of features remains an important area for future research."

- Comparison with other methods: The authors did not compare their method's results (performance/accuracy) with meta-analysis or federated learning. They believe that this "may not be needed":

We believe that the DLMM algorithm achieves a good balance between handling heterogeneity, preserving privacy, accuracy (lossless) and communication efficiency (one round of transferring aggregate data), and comparing to other methods may not be needed.

REVIEWER COMMENTS

Reviewer #1 (Remarks to the Author):

I thank the authors for their answers and revisions. There are two points that I could have brought before (sorry for that).

1. The lossless property may be considered as important. One alternative approach that can be used when no individual patients data (IPD) can be shared is to fit the models on each dataset and then meta-analyze the results. Methods have been studied when IPD are available for one center (or study), but not the others (Debray et al. *Statist Med.* 2012;31:2697-2712, Debray et al. *Statist Med.* 2014;33:2341-2362) or using only aggregated information (Sheng et al. *Statist Med.* 2014;33:2567-2576). The focus in those articles was more on logistic or Cox regression than on linear regression, but I wondered how much loss there would be compared to the proposed DLMM approach. I'm not asking for extensive additional analyses but perhaps some discussion could help.

Response: *We thank the reviewer for these important comments and pointing out these relevant literature. The linear mixed model (LMM) use in our study is closely connected with the (random-effects) meta-analysis. Debray et al. 2012 used the meta-analysis to synthesize the common effects of predictors and further calibrate local IPD prediction. Specifically, if a (random-effects) meta-analysis approach is used, the BLUP for site-specific random effects could also be estimated for the purpose of site-specific prediction. This is very similar to what LMM aims to do. We thus provide a comparison of LMM with (random-effects) meta-analysis approach for the LOS study, as summarized by Supplementary Figure 5. Specifically, we compared three methods:*

- 1) *BLUP (meta): best linear unbiased predictor (BLUP) of intercept or a covariate effect using two-stage IPD-meta-analysis;*
- 2) *Individual LM est: Estimate based on the data from a given site only;*
- 3) *BLUP (LMM): BLUP based on the proposed (one-stage) DLMM algorithm (which is identical to the BLUP based on a LMM).*

The results in the plot below show similar (but not identical) fixed-effects estimates (dotted and dashed horizontal lines), as well as the shrinkage pattern from the individual estimates to the estimated BLUPs based on the BLUP (meta) or BLUP (LMM), for the intercepts (left panel) and effects of age group of 65-80 (right panel). More details are provided in the revised Supplementary Materials. We note that overall the estimated BLUPs and the fixed effects are similar, yet such similarity could be dependent on various factors, such as the number of patients per site, the ratio between the within-site heterogeneity and the between-site heterogeneity (which is corresponding to the intra-cluster correlation), and the number of sites.

Regarding the other literature (Debray et al. 2014; Sheng et al. 2014), they more or less require assuming homogeneity of the associations cross sites (i.e. studies). The advantage of these approaches is the flexibility with missing predictors among sites when integrating data. As a result, their main benefit is on prediction at some site, but not on the synthesis of association effects.

We have added the following discussion to the discussion of the manuscript:

“As suggested by a reviewer, the linear mixed model is closely connected with the (random-effects) meta-analysis, as they both assume the association effects are random and can shrink site-specific (or study-specific) estimation which benefits prediction performance. A comparison of our LMM (or equivalently DLMM) and the random-effects meta-analysis for the LOS study is demonstrated in the Supplementary Materials. The results show that the estimation of common fixed-effects and site-specific random effects (i.e. BLUPs) are similar but not identical. However, such difference depends on various factors, such as the number of patients per site, the ratio between the within-site heterogeneity and the between-site heterogeneity, and the number of sites. Meta-analysis-based model aggregation is extensively studied in literature; for example, see Debray et al 2012; Debray et al 2014; and Sheng et al. 2014 for prediction purposes. A comprehensive comparison between LMM and meta-analysis is however beyond the scope of this paper.”

2. Another issue that could be considered when building DLMMs with interactions between variables is the potential for aggregation bias. This has been raised for testing interactions in meta-analyses of randomized trials using IPDs, including linear models, and some solutions have been proposed (Riley et al. Statist Med. 2020;39:2115-2137). Again, there is no need to implement that, but the issue could be brought to the readers as a caveat in the discussion.

Response: We thank the reviewer for the comments and suggested literature of meta-analyses on interaction effects. Testing interactions between treatment and covariates can help identify which individuals benefit most from particular treatments and thus is an important topic in personalized medicine (Fisher et al. 2017). When integrating IPDs from multiple clinical trials to estimating the interactions, it's important to separate the between-study information from the within-study information, and thus avoid the aggregation bias (Fisher et al. 2017; Riley et al. 2020). This boils down to either including a meta-regression term in the main effect of treatment (models 6-8 in Riley et al. 2020), or assuming the main effect of treatment is common across studies (models 9-11 in Riley et al. 2020). All these models are fitted in the mixed-effect framework. Specifically, for a continuous outcome, the linear mixed models, i.e. models 6 and 9 in Riley et al.

2020, can be conveniently fitted by our proposed DLMM algorithm, when some or all sites (or studies) are not able to share IPDs. Moreover, we are also developing a similar distributed algorithm for fitting mixed-effect models with non-continuous outcomes, see a preprinted paper <https://doi.org/10.1101/2021.05.03.21256561>. This distributed Penalized Quasi Likelihood algorithm is based on DLMM and can be used to test interaction models such as those in Riley et al. 2020.

In the revised manuscript, we have added the following to the discussion

“We also note that in the setting of modeling interactions, caution should be taken in the formulation of the regression model in LMM. Through various proposed formulations that distinct between-study information from the within-study information (Fisher et al. 2017; Riley et al. 2020), the aggregation bias can be avoided.”

Reviewer #2 (Remarks to the Author):

- System/Threat Model: The authors seem to have clarified this issue, and motivate it via the OHDSI network example (which they are part of).

- Model Access: Each party gets the resulting global model in clear. While this allows fairness/reproducibility etc., we know that this model access (white-box) is prone to various privacy attacks.

- Privacy Mechanism: Still, besides simple (cleartext) aggregation, the authors have not incorporated any protection mechanism, e.g., encryption or differential privacy: We mentioned that aggregates leak information about individuals and their response to our criticism is "we might do this in the future" or "we will inspect the aggregates before sharing them" (this can not work without any rigorous analysis). Overall, they push the whole criticism to future work as per their response:

"However, our aggregated data releasing mechanism has not been rigorously studied to meet privacy-preserving criteria such as k-anonymity or differential privacy 27–29. ... In future collaborations using DLMM, we suggest data contributors review the aggregated data to avoid sparse cells (e.g. no cell count less than 5) before sending them to other sites. We will quantify the risk of privacy leaking more rigorously, and enhance our DLMM algorithm via techniques such as differential privacy and multiparty homomorphic encryption³⁰ in the future."

- Scalability Analysis: The authors did not do a scalability analysis with respect to the pxp matrix that is being shared by each party. Again, they leave this as future work:

"Regarding scalability, our DLMM algorithm has great scalability in terms of large number of sites and large number of patients per site. However, in the presence of high dimensional features (i.e., large p), the current algorithm will require sharing of high dimensional matrices, which may be challenging. Extension of DLMM algorithm to improve the scalability on large number of features remains an important area for future research."

Response: *We thank the reviewer for acknowledging our clarification in the previous responses. While we admit the lack of rigorous analysis of privacy protection, we want to point out that the main contribution of this paper is a convenient one-shot lossless algorithm for association study in the biomedical area. The protection of patients' privacy by aggregation, though not rigorously guaranteed, enables many of multi-site collaborations when patient-level data cannot be shared in many of multi-site biomedical studies with an umbrella research IRB, such as the ones at the OHDSI consortium and various PCORnet consortia including PEDSnet (a national pediatric learning health system in United States, Forrest, et al. 2014 JAMIA).*

Regarding the scalability, the communication of $p \times p$ matrix is affordable and remains privacy-preserving as long as the number of predictors p is not very large. This is usually the case in biomedical studies with clinician-driven modeling strategy, as predictors can be chosen by clinical knowledge, such as patients' characteristics related to COVID-19 inpatient treatment. If the dimension of predictors is high, some pre-screening of candidate predictors using IPDs at individual sites could guide the selection of predictors.

- Comparison with other methods: The authors did not compare their method's results (performance/accuracy) with meta-analysis or federated learning. They believe that this "may not be needed":

We believe that the DLMM algorithm achieves a good balance between handling heterogeneity, preserving privacy, accuracy (lossless) and communication efficiency (one round of transferring aggregate data), and comparing to other methods may not be needed.

Response: *We thank the reviewer for emphasizing the importance of comparison with other methods.*

Regarding the comparison with meta-analysis method:

The linear mixed model (LMM) use in our study is closely connected with the (random-effects) meta-analysis. Debray et al. 2012 used the meta-analysis to synthesize the common effects of predictors and further calibrate local IPD prediction. Specifically, if a (random-effects) meta-analysis approach is used, the BLUP for site-specific random effects could also be estimated for the purpose of site-specific prediction. This is very similar to what LMM aims to do. We thus provide a comparison of LMM with (random-effects) meta-analysis approach for the LOS study, as summarized by the Figure below. Specifically, we compared three methods:

- 1) BLUP (meta): best linear unbiased predictor (BLUP) of intercept or a covariate effect using two-stage IPD-meta-analysis;*
- 2) Individual LM est: Estimate based on the data from a given site only;*
- 3) BLUP (LMM): BLUP based on the proposed (one-stage) DLMM algorithm (which is identical to the BLUP based on a LMM).*

The results in the plot below show similar (but not identical) fixed-effects estimates (dotted and dashed horizontal lines), as well as the shrinkage pattern from the individual estimates to the estimated BLUPs based on the BLUP (meta) or BLUP (LMM), for the intercepts (left panel) and effects of age group of 65-80 (right panel). More details are provided in the revised Supplementary Materials. We note that overall the estimated BLUPs and the fixed effects are similar, yet such similarity could be dependent on various factors, such as the number of patients per site, the ratio between the within-site heterogeneity and the between-site heterogeneity (which is corresponding to the intra-cluster correlation), and the number of sites.

We have added the following discussion to the discussion of the manuscript:

“As suggested by a reviewer, the linear mixed model is closely connected with the (random-effects) meta-analysis, as they both assume the association effects are random and can shrink site-specific (or study-specific) estimation which benefits prediction performance. A comparison of our LMM (or equivalently DLMM) and the random-effects meta-analysis for the LOS study is demonstrated in the Supplementary Materials. The results show that the estimation of common fixed-effects and site-specific random effects (i.e. BLUPs) are similar but not identical. However, such difference depends on various factors, such as the number of patients per site, the ratio between the within-site heterogeneity and the between-site heterogeneity, and the number of sites. Meta-analysis-based model aggregation is extensively studied in literature; for example, see Debray et al 2012; Debray et al 2014; and Sheng et al. 2014 for prediction purpose. A comprehensive comparison between LMM and meta-analysis is however beyond the scope of this paper.”

Regarding the comparison with “federated learning”:

*We are not sure about which federated learning approaches should be compared with. We do believe that our DLMM algorithm belongs to the general federated learning framework (with the unique feature of being “one-shot” and lossless) and we do not see the need of comparing with other federated learning methods for the sake of doing comparison. Please kindly advise if you insist on comparing with a **specific** federated learning approach.*

References

1. Debray, T. P. A., Koffijberg, H., Vergouwe, Y., Moons, K. G. M. & Steyerberg, E. W. Aggregating published prediction models with individual participant data: a comparison of different approaches. *Stat. Med.* **31**, 2697–2712 (2012).
2. Debray, T. P. A. *et al.* Meta-analysis and aggregation of multiple published prediction models. *Stat. Med.* **33**, 2341–2362 (2014).
3. Sheng, E., Zhou, X. H., Chen, H., Hu, G. & Duncan, A. A new synthesis analysis method for building logistic regression prediction models. *Stat. Med.* **33**, 2567–2576 (2014).
4. Fisher, D. J., Carpenter, J. R., Morris, T. P., Freeman, S. C. & Tierney, J. F. Meta-analytical methods to identify who benefits most from treatments: daft, deluded, or deft approach? *bmj* **356**, (2017).
5. Riley, R. D. *et al.* Individual participant data meta-analysis to examine interactions between treatment effect and participant-level covariates: statistical recommendations for conduct and planning. *Stat. Med.* **39**, 2115–2137 (2020).
6. Forrest, C. B. *et al.* PCORnet® 2020: current state, accomplishments, and future directions. *J. Clin. Epidemiol.* **129**, 60–67 (2021).

Reviewers' Comments:

Reviewer #1:

Remarks to the Author:

I thank the authors for their thoughtful answers, additional analyses and added discussion.

I have no comment left.

Reviewer #2:

Remarks to the Author:

The authors have addressed most of our concerns. Hereunder are a few comments / suggestions.

Privacy: The authors insist that aggregation is sufficient for this kind of studies (and apparently in the US they can get the necessary permissions). It is unlikely that this would be acceptable under EU GDPR. The authors should include a note about this.

Scalability Analysis: The authors claim that the variable p (i.e., number of features) is small for the kind of studies they are interested in, so scalability is not really their concern (although that would be a problem in other kinds of studies, e.g., genomics). The authors should include a note about this.

Method Comparison: The authors addressed this concern by comparing their method with a meta-analysis technique and while the results are not fabulous they at least did it.

REVIEWERS' COMMENTS

Reviewer #1 (Remarks to the Author):

I thank the authors for their thoughtful answers, additional analyses and added discussion. I have no comment left.

Response: *We thank the reviewer for accepting our last revision.*

Reviewer #2 (Remarks to the Author):

The authors have addressed most of our concerns. Hereunder are a few comments / suggestions.

Privacy: The authors insist that aggregation is sufficient for this kind of studies (and apparently in the US they can get the necessary permissions). It is unlikely that this would be acceptable under EU GDPR. The authors should include a note about this.

Response: *We thank the reviewer for the reminder. Our international data do contain a site from the EU, i.e. the SIDIAP data from Spain, and a site, i.e. the HIRA data from South Korea. Our collaborator confirms that the aggregated data is allowed to be transferred under GDPR. However the data privacy regulations vary across countries. In general, we do agree with the reviewer that the transferring of any aggregated data should be subject to local privacy regulations. We add a note in paragraph 4 of the Discussion:*

"... Also, the privacy regulation of releasing aggregated data could vary across countries and data providers. The disclosure of aggregated data in the DLMM algorithm needs to meet the local privacy requirement. ..."

Scalability Analysis: The authors claim that the variable p (i.e., number of features) is small for the kind of studies they are interested in, so scalability is not really their concern (although that would be a problem in other kinds of studies, e.g., genomics). The authors should include a note about this.

Response: *We thank the reviewer for this note. We agree with the reviewer that the proposed distributed algorithm should be used in a proper setting, i.e. a multi-site association analysis with a small number of predictors and possible heterogeneous association across sites. Studies that involve large number of predictors/features are not suitable for the proposed method as the aggregated data would be too large to be transferred across sites due to privacy concerns. We now add a note in the Discussion:*
"... the current algorithm will require sharing of $p \times p$ dimensional matrices, which may be too large to be transferred across sites due to privacy concerns. As a result, studies that involve a large number of predictors/features (e.g. a large-scale genomic study) are not suitable for the proposed method. ..."

Method Comparison: The authors addressed this concern by comparing their method with a meta-analysis technique and while the results are not fabulous they at least did it.

Response: *We thank the reviewer for agreeing with our last revision.*